# Connecting the Dots: A Chain-of-Collaboration Prompting Framework for LLM Agents

## Abstract

Large Language Models (LLMs) have demonstrated impressive performance in executing complex reasoning tasks. Chain-of-thought effectively enhances reasoning capabilities by unlocking the potential of large models, while multi-agent systems provide more comprehensive solutions by integrating the collective intelligence of multiple agents. However, both approaches face significant limitations. Single-agent with chain-of-thought, due to the inherent complexity of designing cross-domain prompts, faces collaboration challenges. Meanwhile, multi-agent systems consume substantial tokens and inevitably dilute the primary problem, which is particularly problematic in business workflow tasks. To address these challenges, we propose **Cochain**, a collaboration prompting framework that effectively solves the business workflow collaboration problem by combining knowledge and prompts at a reduced cost. Specifically, we construct an integrated knowledge graph that incorporates knowledge from multiple stages. Furthermore, by maintaining and retrieving a prompts tree, we can obtain prompt information relevant to other stages of the business workflow. We perform extensive evaluations of Cochain across multiple datasets, demonstrating that Cochain outperforms all baselines in both prompt engineering and multi-agent LLMs. Additionally, expert evaluation results indicate that the use of a small model in combination with Cochain outperforms GPT-4. The codes are available at https://anonymous.4open.science/r/Cochain-6866.

## 1 Introduction

Large Language Models (LLMs) have demonstrated impressive performance in language understanding and complex reasoning tasks (Touvron et al., 2023a; Bai et al., 2023; GLM et al., 2024; Bi et al., 2024), with prompt engineering playing a crucial role in enhancing their reasoning capabilities. Prompt engineering represented by Chain-of-Thought prompt LLMs in ways that facilitate coherent and step-by-step reasoning processes, endowing agents with depth of thought (Besta et al., 2024; Wei et al., 2022). To enhance agent collaborative capabilities, various multi-agent frameworks have been developed, providing agents with breadth of thought. These frameworks encode carefully designed agent profiles and collaborative mechanisms into prompts, yielding favorable outcomes across domains such as healthcare (Tang et al., 2024; Kim et al., 2024; Nori et al., 2023), education (Dan et al., 2023; Qu et al., 2024; Gu et al., 2024), law (Cui et al., 2024), and finance (Yang et al., 2023). Due to their exceptional problem-solving and collaborative abilities, multi-agent systems are considered a promising pathway toward Artificial General Intelligence (AGI) (Bo et al., 2024; Zhu et al., 2025). However, this raises a critical question: **Is a multi-agent LLM system all you need?**

Evidently not, as both single-agent and multi-agent systems exhibit limitations (Tran et al., 2025). As illustrated in Figure 1, the inherent complexity in designing cross-domain prompts (Sahoo et al., 2024) means that single agents employing prompt engineering lack constraint awareness and face challenges in cross-domain collaboration (under-collaboration). Simultaneously, multi-agent systems face challenges from expensive token consumption and extensive inference time (Du et al., 2024; Wu et al., 2025), with inter-agent communications often ineffectively utilized (Zhang et al., 2024c). More importantly, existing research primarily focuses on maximizing collaboration, extensively exploring how multi-agent systems can improve decision quality (Zhang et al., 2025b; Qian et al., 2025), make safe decisions (Li et al., 2023b; Piatti et al., 2024), and solve problems (Zhou et al., 2024a; Li et al., 2025), while paying insufficient attention to the negative effects of excessive agent collaboration. When all agents simultaneously participate in every decision-making process, core

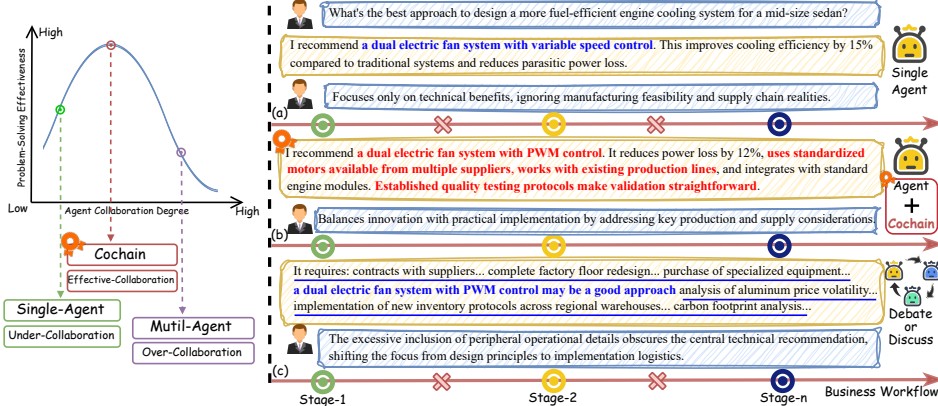

Figure 1: The impact of agent collaboration degree on problem-solving effectiveness. Under-collaboration ignores stage constraints, while over-collaboration dilutes main issues (underlined in examples). Cochain achieve balance by considering constraints while focusing on key problems, effectively connecting the dots. Manual responses provided a brief evaluation of the answers.

issues are frequently overwhelmed by non-critical information, leading not only to a decline in answer quality but also to a preference for consensus-driven answers over accurate ones (Du et al., 2024), particularly in business workflow tasks (Lei et al., 2025). **To our best knowledge, no existing literature has systematically analyzed and defined this "over-collaboration" phenomenon or proposed targeted solutions**. This prompts us to consider: Why not combine the reasoning capabilities of chain-of-thought prompting with the collaborative nature of multi-agent systems?

Inspired by these challenges, we propose a collaborative prompting framework, **Cochain**, designed to enhance LLM agents collaborative capabilities in business workflow tasks at low cost, while mitigating over-collaboration through indirect rather than direct agent participation in decision-making. As shown in Figure 2, Cochain comprises two primary components: collaborative knowledge graph and prompts tree. We mine agent knowledge through counterfactual reasoning and refine it into a knowledge graph integrated with explicit dataset knowledge. To mitigate the impact of irrelevant knowledge on model performance (Zhou et al., 2024b), we introduce causal chain structure to improve cross-stage reasoning capabilities. Inspired by how the human brain integrates fragmented knowledge into coherent thought (Courellis et al., 2024), we propose prompts tree that distills prompts from agent responses. By retrieving from the prompts tree, agents can automatically acquire chain-like cross-stage prompts, avoiding the high cost of manual prompt design while effectively utilizing inter-agent communications. The main contributions are summarized as follows:

- **Effective Collaboration:** We propose an agent collaborative prompting framework, Cochain, that equips agents with both depth and breadth of thinking, effectively alleviating the problems of over-collaboration and under-collaboration.

- **Cost-Effective yet Efficient:** Our extensive experimental results demonstrate that Cochain exhibits significant improvements over all baselines across different model backbones. Cochain possesses reasoning speeds comparable to single-agent systems and even lower inference costs, and surpasses existing multi-agent systems in capability.

- **The Small + Cochain Outperforms the Large:** Evaluations by domain experts have further confirmed that the combination of the small model and our framework outperforms the large model. To support further research on over-collaboration and business workflow tasks, we will publicly release our collaborative knowledge graph.

## 2 RELATED WORK

### 2.1 KNOWLEDGE GRAPH AUGMENTED LLMS

LLM reasoning capabilities are critical for high-quality responses (Jain et al., 2024; Suzgun et al., 2022; Kojima et al., 2022). Knowledge graphs, with their structured, explicit, and interpretable

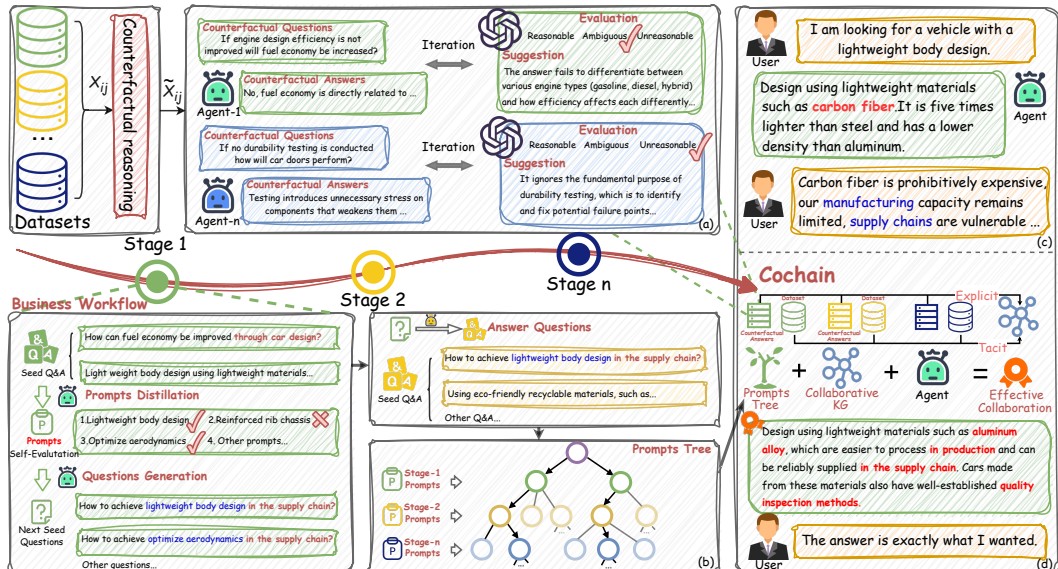

Figure 2: Cochain connects business workflow stages. In part (a), agents address counterfactual inquiries, with answers undergoing interactive iteration. In part (b), a prompts tree is built. Part (d) presents our framework: an agent uses the prompts tree and a collaborative knowledge graph built from counterfactual answers and original data, achieving more collaborative results than part (c).

nature (Zhao et al., 2024b; Pan et al., 2024; Zhou et al., 2025), provide clear representations and transparent reasoning paths, guiding LLMs toward deeper reasoning and mitigating limitations (Ling et al., 2023; Yao et al., 2023b; Wang et al., 2024). For domain-specific LLMs, knowledge graphs enable continuous knowledge updates to accommodate evolving information (Ibrahim et al., 2024; Mariotti et al., 2024; Lavrinovics et al., 2025). Applications include MindMap (Wen et al., 2024) in medicine and ChatLAW (Cui et al., 2024) in law. Business workflows, due to their multi-stage, cross-domain nature, present the unique challenge of multi-stage knowledge fusion. To prevent performance degradation from irrelevant knowledge (Ouyang et al., 2022), we propose a causal chain to optimize knowledge selection, enhancing cross-stage understanding and filtering noise.

## 2.2 PROMPT ENGINEERING

Prompt engineering guides LLMs to maximize their potential via designed prompts (Chen et al., 2023a; Sahoo et al., 2024; Zhao et al., 2023). It enables model adaptation to new tasks through in-context learning and instruction following (Brown et al., 2020; Li et al., 2023a). Well-designed prompts are shown to enhance performance, particularly on complex tasks (Wei et al., 2022; Wang et al., 2023b; Zhou et al., 2023; Yao et al., 2023a). CoT (Wei et al., 2022) enables models to generate intermediate reasoning steps via simple prompts. ToT (Yao et al., 2024) explores coherent text units as intermediate problem-solving steps. CPO (Zhang et al., 2024b) leverages non-optimal reasoning paths from tree search to speed up inference. However, existing methods lack sustained constraint awareness and neglect cross-domain collaboration. Cochain enables LLM agents to address this challenge while balancing interdependent constraints and implementation feasibility.

## 2.3 COLLABORATION OF LLM AGENTS

Many studies in LLM research solve problems through multi-agent collaboration (Akata et al., 2023; Du et al., 2024; Guo et al., 2024; Zhao et al., 2024a; Hao et al., 2023). Common paradigms include discussion (Chen et al., 2024b; Tang et al., 2024; Saha et al., 2024) and debate (Du et al., 2024; Xiong et al., 2023; Chen et al., 2023b) to enhance reasoning, hierarchical structures with specialized roles (Zhang et al., 2025a), and sequential or tree-structured architectures (Zhang et al., 2024c; Zhao et al., 2024a). These have proven effective in domains like medicine (Tang et al., 2024), long-text processing (Zhang et al., 2024c; Zhao et al., 2024a; Sun et al., 2023; Chen et al., 2024a), social

simulation (Zhang et al., 2024a; Wei et al., 2024), and code intelligence (Wang et al., 2023a; Huang et al., 2023). Unlike these approaches, Cochain relies on knowledge fusion rather than token-intensive role-playing. **To the best of our knowledge, we are the first to apply LLM agent collaboration to business workflow tasks**, offering an efficient solution that significantly reduces computational overhead while maintaining strong real-world performance.

## 3 METHODS

### 3.1 COLLABORATIVE KNOWLEDGE GRAPH

The construction of the collaborative knowledge graph relies on a large amount of stage-specific knowledge. We acquire this knowledge from the training sets of the datasets, which are represented as $D_i = \{(X_{ij}, Y_{ij})\}_{j=1}^{n_i}$ where $X_{ij} \in X_i$ is the input and $Y_{ij} \in Y_i$ is the corresponding output. The datasets are subject to data cleaning and triplet extraction to acquire knowledge for constructing the explicit knowledge graph $\mathcal{KG}_{\text{explicit}}$:

$$\mathcal{KG}_{\text{explicit}} = \bigcup_{i=1}^{N} \bigcup_{j=1}^{n_i} \text{ExtractTriples}(X_{ij}, Y_{ij}) \tag{1}$$

In order to leverage the capabilities of other business workflow agents, we need to identify the tacit variables that play a critical role in the agent's responses. Since extracting the internal tacit variables of agents is challenging, we focus on the extraction of tacit knowledge from the hidden layers. To achieve this, we introduce counterfactual reasoning. Specifically, we perform causal reasoning, adversarial reasoning, substitution reasoning, extreme counterfactual reasoning, and backward causal reasoning on the questions within the datasets. For each sample $(X_{ij}, Y_{ij})$, we generate the corresponding counterfactual input $\tilde{X}_{ij}$:

$$\tilde{X}_{ij} = \text{GenerateCounterfactual}(X_{ij}) \tag{2}$$

We then input these questions into the relevant vertical domain agents to generate answers. To represent the output of tacit knowledge influenced by tacit variables, we introduce the tacit variable $\theta_{ij}$, which represents the tacit knowledge or state inside the model and plays a key role in generating counterfactual outputs. The tacit knowledge $h_{ij}$ is the model's hidden state under the influence of the tacit variable $\theta_{ij}$, which acts as an intermediary in the process of generating counterfactual outputs. The tacit knowledge $h_{ij}$ is influenced by the tacit variable $\theta_{ij}$ where $h_{ij} \mid \theta_{ij} \sim P(h_{ij} \mid \theta_{ij})$. The tacit knowledge $h_{ij}$ further influences the generation of the counterfactual output $\tilde{Y}_{ij}$, where $\tilde{Y}_{ij} \mid h_{ij}, \tilde{X}_{ij} \sim P(\tilde{Y}_{ij} \mid h_{ij}, \tilde{X}_{ij})$. The final generation process is expressed as (Appendix B):

$$P(\tilde{Y}_{ij} \mid \tilde{X}_{ij}) = \int_{\Theta} P(\tilde{Y}_{ij} \mid h_{ij}, \tilde{X}_{ij}) P(h_{ij} \mid \theta_{ij}) P(\theta_{ij} \mid \tilde{X}_{ij}) d\theta_{ij} \tag{3}$$

In this process, $P(\tilde{Y}_{ij} \mid h_{ij}, \tilde{X}_{ij})$ represents the probability of generating the counterfactual output given the tacit knowledge $h_{ij}$ and the counterfactual input $\tilde{X}_{ij}$. $P(h_{ij} \mid \theta_{ij})$ represents the generative distribution of the tacit knowledge $h_{ij}$ given the tacit variable $\theta_{ij}$, and $P(\theta_{ij} \mid \tilde{X}_{ij})$ represents the prior distribution of the tacit variable $\theta_{ij}$ given the counterfactual input $\tilde{X}_{ij}$. Subsequently, these answers are evaluated by a general-purpose LLM, and the evaluation results are classified into three levels: reasonable, ambiguous, and unreasonable. The general-purpose model provides feedback on evaluation opinions and suggestions, which are then used by the vertical domain agent to generate another response. This process iterates until a reasonable answer is obtained. During knowledge distillation, we apply causal enhancement, with special attention to causal keywords such as "depends on", "relies on", and "applies to". Using the estimated tacit variable $\theta_{ij}$, tacit knowledge is extracted from the counterfactual input-output pairs $(\tilde{X}_{ij}, \tilde{Y}_{ij})$ completed through iteration, and a tacit knowledge graph $\mathcal{KG}_{\text{tacit}}$ is constructed:

$$\mathcal{KG}_{\text{tacit}} = \bigcup_{i=1}^{N} \bigcup_{j=1}^{n_i} \text{ExtractTriples}(\tilde{X}_{ij}, \tilde{Y}_{ij}, \theta_{ij}) \tag{4}$$

Finally, by integrating the explicit and the tacit knowledge graph, we construct a collaborative knowledge graph $\mathcal{KG}$ that contains both stage-specific knowledge and inter-stage connectivity:

$$\mathcal{KG} = \mathcal{KG}_{\text{explicit}} \cup \mathcal{KG}_{\text{tacit}} \tag{5}$$

## 3.2 CAUSAL CHAIN

To enable the agent to organize the causal relationships between various knowledge stages and generate more accurate and coherent responses, rather than simply listing knowledge, we introduce a causal chain mechanism. When a user submits a request to the agent, the request is first decomposed, and keywords are extracted. These keywords are then input into the knowledge graph to match relevant nodes.

Specifically, as shown in Figure 3, we use a pre-trained text encoder (Reimers & Gurevych, 2019) to convert the extracted keywords and their related nodes into vector representations. For the vector representation $v_k$ of keyword $k$ and the vector representation $v_n$ of node $n$ in the knowledge graph, we compute the cosine similarity between these vectors to identify the most relevant stage knowledge:

$$\text{Sim}(v_k, v_n) = \frac{v_k \cdot v_n}{\|v_k\|\|v_n\|} \tag{6}$$

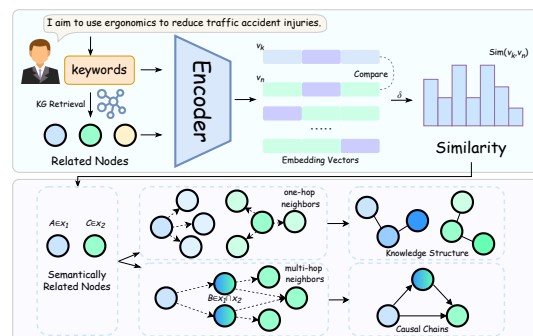

Figure 3: Retrieving relevant knowledge from the collaborative knowledge graph and the process of constructing causal chains.

By setting a similarity threshold $\delta$, we select knowledge nodes that are highly related to the semantics of the keywords. For these nodes, we extend their one-hop neighbors to introduce additional associated knowledge and construct a triple-based knowledge structure to enhance the agent's understanding capability. To address the challenge of cross-stage integration, we adopt a multi-hop neighbor exploration method to build causal chains. By leveraging stage-specific knowledge as a bridge, this approach effectively explains inter-stage knowledge relationships. Specifically, within the stage set $X = \{x_1, x_2, x_3, \dots\}$, for a node $A$ with high semantic similarity, where $A \in x_1$, we expand its one-hop neighbor node $B$, where $B \in x_1 \cap x_i$ $(i \neq 1)$, and use this as a bridge to explore nodes $C$ from other stages, where $C \in x_i$ $(i \neq 1)$, to construct the causal chain:

$$\text{CausalChain}(A, C) = A \xrightarrow{B} C \tag{7}$$

Using the causal chain mechanism, the agent is able to comprehend and identify the causal relationships between the knowledge we provide. This enables the agent to consider the interdependencies between different components when generating responses to user queries.

## 3.3 PROMPTS TREE

In business workflow tasks, the interconnections between various stages are crucial for effective collaboration. To achieve this, we propose an innovative prompt distillation method aimed at constructing a prompts tree. This method derives stage-specific Q&A from the seed Q&A using a fragmentation approach, where each node at every level of the tree corresponds to prompts from different stages. By querying the prompts tree, it is possible to obtain prompt information that spans the entire business workflow. Specifically, we start with seed Q&A in Stage 1 and utilize the agent from Stage 1 to distill solution-oriented prompts from the answers. The most effective $m$ prompts are selected through self-evaluation. These $m$ prompts are then used to generate new question templates, focusing primarily on how to implement or solve the prompts distilled from the previous stage in the next stage. Each question is answered, forming a new seed Q&A. These Q&A are further involved in the prompt distillation of the current stage, driving the content to extend to the next stage.

In this process, the user need is treated as the root node, with prompts in Stage 1 derived as child nodes based on the user need. These Stage 1 prompts further generate Q&A in Stage 2, from which Stage

Table 1: Performance comparison of different model backbones and multi-agent baselines across three business domains. We report the mean and standard error (SE) over five experiments. BERTScore F1 is abbreviated as BS-F. We highlight the best and second-best results.

| Backbone | Baseline | Automotive | | | Pharmaceutical | | | E-commerce | | |
|---|---|---|---|---|---|---|---|---|---|---|
| | | BS-F | GLEU | ROUGE-L | BS-F | GLEU | ROUGE-L | BS-F | GLEU | ROUGE-L |
| Qwen2-7B | PMC | 65.86±0.03 | 13.23±0.08 | 18.35±0.07 | 66.97±0.79 | 16.25±0.58 | 24.75±0.62 | 71.02±0.17 | 24.18±0.37 | 29.93±0.45 |
| | MedAgents | 65.16±0.25 | 12.37±0.17 | 18.33±0.18 | 65.84±0.15 | 10.68±0.36 | 20.83±0.50 | 69.45±0.19 | 17.42±0.17 | 33.38±0.18 |
| | Debate(short) | 65.30±0.26 | 12.42±0.37 | 17.63±0.23 | 70.62±0.41 | 16.04±0.35 | 24.12±0.49 | 73.03±0.43 | 20.89±0.32 | 28.27±0.29 |
| | Debate(long) | 65.61±0.27 | 12.85±0.47 | 18.28±0.13 | 60.57±0.30 | 10.28±0.39 | 11.21±0.52 | 72.41±0.60 | 20.82±0.59 | 28.46±0.53 |
| | CoA | 70.70±0.53 | 21.51±0.41 | 22.65±0.68 | 76.98±0.44 | 30.45±0.42 | 34.00±0.17 | 79.24±0.18 | 38.60±0.35 | 39.14±0.62 |
| | Cochain | 75.05±0.17 | 27.66±0.26 | 28.43±0.23 | 78.50±0.04 | 34.48±0.28 | 35.23±0.17 | 80.22±0.16 | 37.83±0.38 | 40.84±0.41 |
| DeepSeek-R1-7B | PMC | 65.94±0.13 | 10.75±0.23 | 15.78±0.13 | 68.06±0.27 | 22.51±0.29 | 28.07±0.20 | 66.66±0.43 | 15.80±0.39 | 23.62±0.30 |
| | MedAgents | 65.58±0.60 | 8.03±0.55 | 14.08±0.59 | 70.35±0.14 | 24.47±0.19 | 30.90±0.08 | 66.98±0.49 | 13.67±0.49 | 20.31±0.60 |
| | Debate(short) | 65.93±0.23 | 11.41±0.44 | 17.33±0.52 | 71.46±0.31 | 27.13±0.27 | 30.34±0.44 | 70.71±0.29 | 23.01±0.22 | 27.02±0.39 |
| | Debate(long) | 66.06±0.67 | 12.13±0.49 | 17.70±0.26 | 71.54±0.42 | 27.34±0.39 | 30.04±0.39 | 70.48±0.39 | 23.07±0.10 | 27.00±0.44 |
| | CoA | 70.83±0.29 | 13.27±0.58 | 17.51±0.58 | 77.06±0.15 | 38.18±0.49 | 35.53±0.23 | 80.43±0.32 | 43.80±0.39 | 42.03±0.43 |
| | Cochain | 73.20±0.02 | 21.45±0.17 | 23.89±0.13 | 79.11±0.09 | 38.27±0.17 | 37.21±0.21 | 82.68±0.05 | 50.19±0.13 | 47.07±0.13 |
| Claude-3.5-haiku | PMC | 64.06±0.04 | 9.73±0.56 | 10.12±0.31 | 66.18±0.16 | 13.04±0.36 | 13.91±0.18 | 66.85±0.07 | 15.98±0.60 | 15.88±0.21 |
| | MedAgents | 64.38±0.10 | 10.77±0.56 | 11.02±0.14 | 66.01±0.17 | 14.60±0.30 | 14.66±0.29 | 68.00±0.13 | 17.58±0.31 | 16.51±0.14 |
| | Debate(short) | 64.36±0.17 | 11.21±0.39 | 13.72±0.33 | 69.11±0.11 | 14.53±0.26 | 16.03±0.28 | 68.41±0.43 | 19.97±0.32 | 18.37±0.31 |
| | Debate(long) | 65.13±0.08 | 11.79±0.49 | 12.13±0.39 | 67.68±0.10 | 15.94±0.31 | 15.13±0.32 | 67.49±0.67 | 17.23±0.42 | 18.98±0.56 |
| | CoA | 63.78±0.26 | 9.23±0.29 | 10.28±0.20 | 66.00±0.39 | 13.33±0.22 | 14.25±0.36 | 67.03±0.35 | 15.02±0.33 | 15.12±0.35 |
| | Cochain | 69.66±0.03 | 18.62±0.06 | 16.83±0.24 | 72.83±0.06 | 23.47±0.13 | 19.25±0.22 | 71.56±0.12 | 20.11±0.13 | 20.65±0.17 |

2-specific prompts are distilled as child nodes of the Stage 1 prompts. In this way, we continuously build and maintain a prompts tree. By retrieving the prompts tree, we automatically generate a prompt chain that covers the entire business workflow, thus avoiding the difficulties associated with manually constructing prompts. These prompts will guide large language models to generate responses that comprehensively consider the content of each stage, ensuring high collaboration across the stages.

# 4 EXPERIMENTS

In our experiments, we evaluate our collaboration framework and answer the following questions: **(1)** To what extent does Cochain enhance inter-agent collaboration capabilities? **(2)** How does Cochain affect agent performance across specialized and cross-domain collaboration tasks? **(3)** Which key design elements of Cochain are most critical for enhancing collaboration capabilities?

## 4.1 EXPERIMENTS SETUP

**Datasets.** Cochain is evaluated on two benchmarks across four datasets: Auto-SLURP (Shen & Shen, 2025), for assessing the end-to-end workflow of multi-agent frameworks, and MSCoRe (Lei et al., 2025), for evaluating multi-stage reasoning and collaboration. **It is worth noting that the automotive workflow test set is rigorously reviewed and optimized by domain experts from a major automobile manufacturer to ensure its practical industry relevance.**

**Metrics.** BERTScore (Zhang et al., 2019) utilizes contextual word embeddings to evaluate semantic similarity, while GLEU (Wu et al., 2016) emphasizes grammatical and semantic quality, and ROUGE-L (Lin, 2004) assesses deep semantic alignment, particularly for extended texts. We employ these metrics as primary indicators of semantic similarity between generated and reference answers. For comprehensive experimental results across additional metrics, refer to Appendix D.

**LLMs.** We utilize sixteen LLMs as the backbone of Cochain. With both proprietary and open-source models, we examine the differences between reasoning-capable models (DeepSeek-R1-7B (DeepSeek-AI, 2025)) and standard models (Qwen2-7B (qwe, 2024)). We also investigate using identical versus mixed models (Llama2-7B (Touvron et al., 2023b), GLM4-9B, Qwen2-7B, DeepSeek-7B) across workflow stages. All open-source models undergo domain-specific fine-tuning to serve as specialized agents in target domains. For proprietary models, we assess GPT-3.5-turbo, GPT-4o (Achiam et al., 2023), DeepSeek-V3.1 (DeepSeek-AI, 2024), and Claude-3.5-haiku (The) performance.

**Baselines.** Our baseline selection encompasses the predominant multi-agent collaboration methodologies (Appendix C). Specifically, PMC (Zhang et al., 2025a) implements multi-agent collaboration through hierarchical planning. MedAgents (Tang et al., 2024) facilitates collaborative decision-making via discussion and voting mechanisms. Debate (Du et al., 2024) enables collaboration through argumentative discourse. CoA (Zhang et al., 2024c) establishes chain-based collaboration.


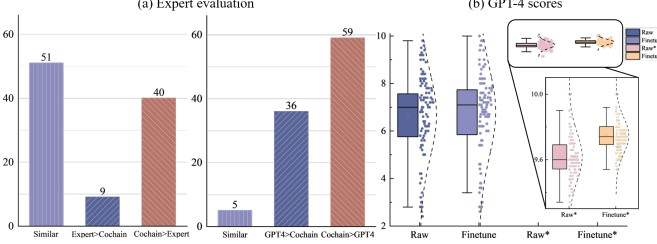

Figure 4: (a) Expert evaluation & (b) GPT-4 scores.

Figure 5: Reasoning time differential.

Table 2: The performance of GPT-3.5-turbo and GPT-4o when using CoT, ToT, and Cochain.

| Backbone | Method | BS-F | GLEU | ROUGE-L |
|---|---|---|---|---|
| GPT-3.5-turbo | + IO | 72.84±0.07 | 15.09±0.17 | 21.98±0.23 |
| | + CoT | 71.61±0.17 | 11.15±0.17 | 17.92±0.16 |
| | + ToT | 66.69±0.05 | 10.08±0.04 | 17.74±0.09 |
| | + Cochain | **76.94±0.03** | **28.92±0.12** | **30.65±0.09** |
| GPT-4o | + IO | 71.92±0.08 | 23.05±0.17 | 23.79±0.17 |
| | + CoT | 72.57±0.15 | 23.76±0.14 | 24.15±0.28 |
| | + ToT | 66.23±0.17 | 12.51±0.14 | 17.52±0.19 |
| | + Cochain | **74.46±0.07** | **25.16±0.25** | **25.07±0.14** |

Table 3: Result of different skip stages on evaluation metrics.

| Skip | Stage | BS-F | GLEU | ROUGE-L |
|---|---|---|---|---|
| zero | / | 75.39 | 28.17 | 28.87 |
| one | $S_1 \rightarrow S_3 \rightarrow S_5$ | 74.00 | 24.66 | 25.18 |
| | $S_2 \rightarrow S_4$ | 74.52 | 25.57 | 26.18 |
| | $S_3 \rightarrow S_5$ | 74.18 | 25.45 | 25.95 |
| two | $S_1 \rightarrow S_3$ | 74.27 | 25.26 | 25.74 |
| | $S_2 \rightarrow S_4$ | 74.07 | 25.00 | 25.82 |

Besides, we compare the performance of single-agent prompt engineering methods, namely CoT (Wei et al., 2022) and ToT (Yao et al., 2024).

## 4.2 ENHANCING COLLABORATIVE REASONING CAPABILITIES IN AGENTS WITH COCHAIN

**Overall Results of Cochain.** Table 1 demonstrates Cochain's performance on business workflow tasks across three diverse language models. Results show Cochain consistently outperforms all baselines by significant margins across all datasets. Table 4 shows that Cochain achieves higher accuracy on the Auto-SLURP bench-

Table 4: Accuracy comparison on Auto-SLURP benchmark.

| Backbone | Baselines | | | | Cochain |
|---|---|---|---|---|---|
| | PMC | MedAgents | Debate | CoA | |
| DeepSeek-V3.1 | 0.12 | 0.07 | 0.32 | 0.27 | **0.33** |
| Claude-3.5-haiku | 0.29 | 0.26 | 0.31 | 0.37 | **0.38** |

mark. Cochain also exhibits substantial advantages in computational efficiency, with shorter reasoning times, fewer output tokens, and lower operational costs compared to other baselines (Appendix D.1). Figure 7 reports consistent performance gains achieved by Cochain as model parameters increase, demonstrating its strong scalability potential with more powerful foundation models.

**Comparative Analysis of Cochain Against Prompting Engineering Approaches.** Table 2 presents a comparative evaluation of Cochain against established prompting methodologies. The results show that while CoT and ToT approaches enhance reasoning capabilities, they exhibit limited collaborative advantages compared to our framework. Specifically, these traditional techniques focus on isolated reasoning paths, whereas Cochain enables agents to incorporate both upstream and downstream factors into decision-making, yielding more comprehensive solutions for complex, interdependent problems across multiple domains of expertise.

**Qualitative Evaluation.** Given the absence of established metrics to evaluate the collaboration of LLM agents, we use a two-fold validation. First, we recruit five experts from a major automotive manufacturer. These experts judge Qwen2-7B with Cochain to outperform GPT-4 in approximately 60% of cases, and to achieve performance on par with or even superior to human experts in approximately 90% of cases (Figure 4(a)). We also employ GPT-4 for quantitative assessment across five scoring rounds, comparing performance with and without fine-tuning and Cochain. Figure 4(b) shows models enhanced with Cochain (Raw* and Finetune*) demonstrate higher median scores, with the fine-tuned version achieving nearly optimal performance (median >9.7).

Table 5: Results of multi-stage collaboration performance on Automotive datasets, highlighting fine-tuned results and Cochain application, and improvements(↑). The experimental results reflect two different scenarios in the workflow. †The score is derived from GPT-4's rating of the final solution (10-point scale), which is synthesized by integrating the answers from each stage.

| Stage | Method | Backbone | BS-F | GLEU | ROUGE-L | Backbone | BS-F | GLEU | ROUGE-L |
|---|---|---|---|---|---|---|---|---|---|
| $S_1$ | Finetuned | | 71.60 | 22.59 | 24.51 | Llama2-7B | 70.68 | 18.79 | 21.49 |
| | + Cochain | | 74.65 | 26.28 | 25.97 | | 71.46 | 20.96 | 21.58 |
| $S_2$ | Finetuned | | 70.44 | 19.37 | 21.70 | GLM4-9B | 69.34 | 16.30 | 19.76 |
| | + Cochain | Qwen2-7B | 74.65 | 26.57 | 25.74 | | 73.19 | 22.33 | 24.06 |
| $S_3$ | Finetuned | | 70.23 | 19.39 | 22.39 | Qwen2-7B | 70.28 | 18.95 | 22.38 |
| | + Cochain | | 74.84 | 27.39 | 26.29 | | 74.51 | 25.76 | 26.00 |
| $S_4$ | Finetuned | | 69.42 | 18.71 | 21.90 | DeepSeek-7B | 68.96 | 13.06 | 18.70 |
| | + Cochain | | 75.39 | 29.28 | 27.94 | | 71.03 | 14.39 | 18.89 |
| | Score† | | 6.3 | 9.2 | +2.9 | | 6.5 | 9.6 | +3.1 |

## 4.3 Cochain Efficacy Across Specialized and Cross-Domain Tasks

To investigate Cochain's impact on specialized and cross-domain tasks, we experimented on automotive industry data using two approaches: a multi-domain model (single Qwen2-7B fine-tuned on all domains) and specialized models (different models each fine-tuned for specific domains), testing on 4 sub-datasets. Table 5 shows Cochain significantly improves both approaches, with particularly strong GLEU score gains. The specialized configuration demonstrates Cochain's ability to orchestrate effective collaboration among domain-expert agents. GPT-4 evaluation scores, assessing comprehensive solutions resulting from four stages, confirm substantial quality improvements for both approaches. Further experiments in Appendix D.3 with specialized models of the same architecture also validate Cochain's effectiveness in coordinating expert agents toward coherent solutions.

## 4.4 Key Design Factors and Hyperparameter Influence in Cochain

**Ablation Study.** To validate Cochain's effectiveness, we conduct ablation experiments with three variants: (1) w/o Knowledge Graph; (2) w/o Causal Chain; and (3) w/o Prompts Tree. As shown in Table 6, all variants exhibit performance degradation across all metrics, confirming each component's importance. Notably, removing the prompts tree causes the most significant drop, highlighting its critical role in collaboration(Appendix D.4). The knowledge graph and causal chain components also prove essential, demonstrating their complementary functions in the reasoning process.

**Hyper-parameter Sensitivity Analysis.** We investigated key hyperparameters by fixing the number of nodes ($n$) from the knowledge graph while varying the prompt count. Figure 6 shows that for any fixed $n$, performance follows an inverted U-shaped curve, indicating an optimal range rather than a monotonic relationship. When fixing prompt count and varying $n$, no consistent pattern emerged, suggesting prompt design exerts greater influence than knowledge quantity, aligning with our ablation

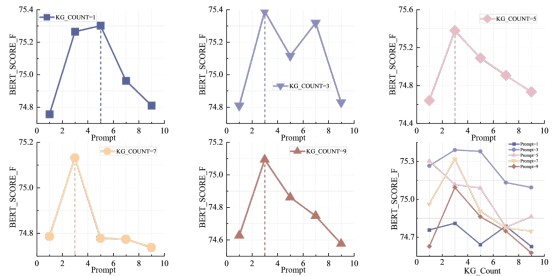

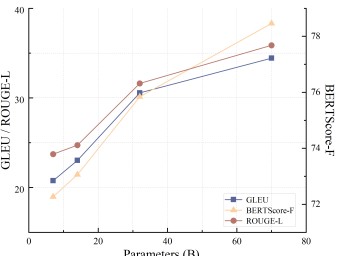

Figure 6: Parameter analysis results. In the first five plots, the top-n value is fixed, while in the last plot, the number of prompts is fixed.

Figure 7: Comparative analysis of metrics scaling with increasing model size from DeepSeek-R1 7B to 70B.

Table 6: Result of ablation study. We highlight the most and second-most efficient modules.

| Backbone | Method | Automotive | | | Pharmaceutical | | | E-commerce | | |
|---|---|---|---|---|---|---|---|---|---|---|
| | | BS-F | GLEU | ROUGE-L | BS-F | GLEU | ROUGE-L | BS-F | GLEU | ROUGE-L |
| Qwen2-7B | Cochain | 75.39 | 28.17 | 28.87 | 78.48 | 33.98 | 35.03 | 79.91 | 37.08 | 40.06 |
| | w/o $\mathcal{KG}$ | 75.21 | 27.81 | 28.74 | 76.34 | 25.92 | 28.09 | 76.72 | 32.40 | 32.74 |
| | w/o Causal Chain | 75.20 | 27.55 | 28.42 | 76.20 | 25.38 | 27.76 | 76.86 | 33.30 | 32.93 |
| | w/o Prompts Tree | 73.08 | 22.45 | 24.56 | 76.19 | 26.84 | 28.91 | 76.59 | 31.83 | 32.38 |
| DeepSeek-R1-7B | Cochain | 73.19 | 21.73 | 24.01 | 78.97 | 37.94 | 36.75 | 82.68 | 50.30 | 47.24 |
| | w/o $\mathcal{KG}$ | 72.41 | 19.71 | 22.63 | 78.22 | 37.33 | 36.40 | 77.36 | 36.54 | 35.85 |
| | w/o Causal Chain | 72.67 | 19.96 | 22.86 | 78.41 | 37.36 | 36.74 | 77.60 | 36.30 | 35.74 |
| | w/o Prompts Tree | 72.27 | 16.54 | 20.60 | 76.96 | 37.98 | 35.61 | 79.02 | 41.03 | 38.84 |
| Claude-3.5-haiku | Cochain | 69.61 | 18.73 | 17.30 | 72.87 | 23.59 | 19.67 | 71.36 | 20.15 | 20.78 |
| | w/o $\mathcal{KG}$ | 68.98 | 16.63 | 16.31 | 70.93 | 16.42 | 18.45 | 71.13 | 18.85 | 20.22 |
| | w/o Causal Chain | 69.33 | 17.31 | 16.65 | 71.33 | 17.43 | 18.78 | 71.23 | 18.89 | 20.45 |
| | w/o Prompts Tree | 65.79 | 12.84 | 13.71 | 70.85 | 16.24 | 18.31 | 69.76 | 15.55 | 18.59 |

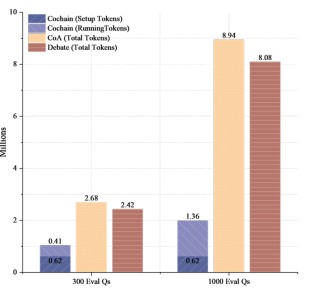
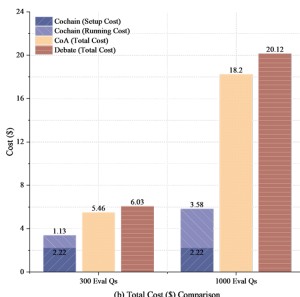
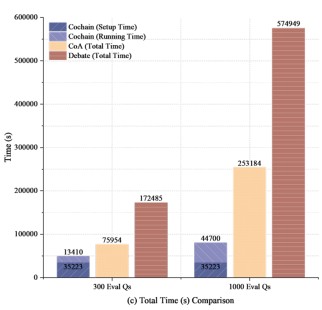

Figure 8: Cost comparison (Tokens, API Fees, Time) across baselines on pharmaceutical dataset. Cochain's total costs include the one-time Prompt Tree setup and costs for varying evaluation queries.

results. Table 3 demonstrates that skipping any stage ($S_k$) in the prompts tree consistently reduces performance, confirming each stage's indispensable role in the collaborative reasoning process.

### 4.5 EFFICIENCY AND COST ANALYSIS

Figure 5 shows that as reasoning rounds increase, the time gap between thinking and non-thinking models grows dramatically for PMC but remains moderate for Cochain, demonstrating Cochain's superior efficiency as complexity increases. Furthermore, as shown in Figure 8, we present Cochain's initial setup overhead as a one-time investment for cost amortization. Our analysis confirms this strategy's value: traditional multi-agent systems have no setup cost but high per-inference costs, whereas Cochain's is substantially lower. Consequently, with more use, Cochain's initial investment is quickly amortized, leading to significant long-term advantages in total cost and time. A further cost analysis is presented in Appendix D.1.

## 5 CONCLUSION

In this paper, we propose Cochain, a chain-of-collaboration prompting framework designed to enhance the collaboration of LLM agents in workflow tasks. Cochain combines the advantages of both single-agent and multi-agent systems, achieving depth in thought and breadth in collaboration. It is characterized by low operational cost, high inference speed, strong interpretability, and robust reasoning chain resilience. Cochain introduces a prompts tree structure for reasoning, enabling collaborative inference paths among multiple agents on-chain. By constructing a cross-node tacit knowledge graph, it improves relational reasoning and reduces hallucinations during collaboration. Additionally, the causal chain mechanism enhances interpretability. Extensive experiments demonstrate that Cochain yields effective improvements in scenarios suffering from both undercollaboration and overcollaboration, excelling in specialized and cross-domain tasks. Expert evaluations also indicate that combining smaller models with Cochain surpasses GPT-4 in workflow tasks.

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

## SUMMARY OF THE APPENDIX

This appendix contains additional details for the paper. The appendix is organized as follows:

- § A describes algorithms and examples to better understand Cochain.
- § B describes the probabilistic formulations and mathematical proof for modeling an agent's unobservable tacit knowledge through the generation of counterfactual outputs.
- § C introduces more details on Cochain.
- § D presents more experiments and analysis on Cochain.
- § E provides a qualitative analysis to interpret the outputs of Cochain, alongside a comparative case study against baseline methods.
- § F presents the prompt templates for Cochain and other baselines.

## A  ALGORITHMS AND EXAMPLES TO BETTER UNDERSTAND COCHAIN

---

**Algorithm 1** Collaborative Knowledge Graph Construction

---

**Require:** Datasets $D = \{D_1, \ldots, D_N\}$; LLM agents $A = \{A_1, \ldots, A_N\}$; Evaluation LLM $E$; Stage Labels $L = \{L_1, \ldots, L_N\}$.
**Ensure:** A collaborative knowledge graph $\text{KG}_{\text{collab}}$.
 1: **Initialize:** $\text{KG}_{\text{explicit}} \leftarrow \emptyset$, $\text{KG}_{\text{tacit}} \leftarrow \emptyset$

 2:                                            ▷ **Phase 1: Build Explicit Knowledge Graph**
 3: **for** $i \leftarrow 1$ **to** $N$ **do**
 4:     **for all** $(\text{instruction}, \text{response}) \in D_i$ **do**
 5:         $\text{triples} \leftarrow \text{ExtractTriples}(\text{instruction}, \text{response})$
 6:         $\text{LabelNodesInTriples}(\text{triples}, L_i)$
 7:         $\text{KG}_{\text{explicit}} \leftarrow \text{KG}_{\text{explicit}} \cup \text{triples}$
 8:     **end for**
 9: **end for**

10:                                            ▷ **Phase 2: Build Tacit Knowledge Graph**
11: **for** $i \leftarrow 1$ **to** $N$ **do**
12:     **for all** $(\text{instruction}, \text{response}) \in D_i$ **do**
13:         $\text{cf\_instruction} \leftarrow \text{GenerateCounterfactual}(\text{instruction})$
14:         **repeat**
15:         $\text{cf\_response} \leftarrow A_i.\text{query}(\text{cf\_instruction})$
16:         $(\text{evaluation}, \text{feedback}) \leftarrow E.\text{evaluate}(\text{cf\_response})$
17:         **if** $\text{evaluation} \neq$ "reasonable" **then**
18:             $\text{cf\_instruction.append}(\text{feedback})$
19:         **end if**
20:         **until** $\text{evaluation} =$ "reasonable"
21:         $\text{tacit\_triples} \leftarrow \text{ExtractTriples}(\text{cf\_instruction}, \text{cf\_response})$
22:         $\text{LabelNodesInTriples}(\text{tacit\_triples}, L_i)$
23:         $\text{KG}_{\text{tacit}} \leftarrow \text{KG}_{\text{tacit}} \cup \text{tacit\_triples}$
24:     **end for**
25: **end for**

26:                                                ▷ **Phase 3: Merge Graphs**
27: $\text{KG}_{\text{collab}} \leftarrow \text{Merge}(\text{KG}_{\text{explicit}}, \text{KG}_{\text{tacit}})$
28: **return** $\text{KG}_{\text{collab}}$

---

---

**Algorithm 2** Causal Chain Retrieval and Construction

---

**Require:** User query query; Stage-labeled knowledge graph $KG_{collab}$; Text encoder Encoder; Seed nodes count top_n; Chain length limit max_depth.
**Ensure:** A textual causal chain causal_chain_text.

1:                                   ▷ **Phase 1: Two-Stage Seed Node Retrieval**
2: keywords ← ExtractKeywords(query)
3: query_vector ← Encoder.**encode**(query)
4: candidate_nodes ← $KG_{collab}$.**KeywordSearch**(keywords)
5: candidate_vectors ← Encoder.**encode**(candidate_nodes)
6: similarities ← **CosineSimilarity**(query_vector, candidate_vectors)
7: seed_nodes ← **GetTopN**(candidate_nodes, similarities, top_n)

8:                                 ▷ **Phase 2: Stage-Aware Multi-hop Expansion**
9: all_paths ← ∅
10: **for** each start_node in seed_nodes **do**
11:                      ▷ Search paths crossing stages via bridge nodes with multiple labels
12:      paths ← $KG_{collab}$.**FindCrossStagePaths**(start_node, min_depth ← 2, max_depth)
13:      all_paths ← all_paths ∪ paths
14: **end for**

15:                                          ▷ **Phase 3: Format and Output**
16: causal_chain_text ← **VerbalizePaths**(all_paths)
17: **return** causal_chain_text

---

**Algorithm 3** Prompts Tree Construction

---

**Require:** Workflow stages $S = \{S_1, \ldots, S_n\}$; Domain agents $A = \{A_1, \ldots, A_n\}$; Seed Q-A pair InitialSeedQA; Best prompts count $m$.
**Ensure:** A prompt tree PromptsTree.
1: **Initialize:**
2: rootNode ← **CreateNode**(InitialSeedQA.question)
3: PromptsTree ← **InitializeTree**(rootNode)
4: queue ← [(rootNode, InitialSeedQA.answer, $S_1$)]
5: **while** queue ≠ ∅ **do**
6:      parentNode, currentAnswer, currentStage ← queue.pop_front()
7:                                 ▷ Phase 1: Distill Prompts
8:      distilledPrompts ← $A_{currentStage}$.**DistillPrompts**(currentAnswer)
9:                                 ▷ Phase 2: Select Best Prompts
10:      bestPrompts ← $A_{currentStage}$.**SelfEvaluate**(distilledPrompts, $m$)
11:                                  ▷ Phase 3: Grow Tree
12:      **if** currentStage ≠ $S_n$ **then**               ▷ Proceed if not the last stage
13:          nextStage ← $S_{index(currentStage)+1}$
14:          **for** each promptText in bestPrompts **do**
15:              childNode ← **CreateNode**(promptText)
16:              parentNode.**add_child**(childNode)
17:              newQuestion ← **GenerateQuestionForNextStage**(promptText, nextStage)
18:              newAnswer ← $A_{nextStage}$.**query**(newQuestion)
19:              queue.**push_back**((childNode, newAnswer, nextStage))
20:          **end for**
21:      **end if**
22: **end while**
23: **return** PromptsTree

---

As shown in the Figure 10, after applying Cochain, the agent anticipates that carbon fiber requires complex molding processes, completely different from traditional metal stamping, and that qualified suppliers are scarce, thus requiring consideration of supply chain stability and high procurement

costs significantly impacting vehicle pricing and profit margins. The prompts tree anticipates existing production processes, guiding the agent to consider compatibility constraints between new equipment and existing stamping-welding lines. Based on this, Cochain generates superior answers compared to single agents.

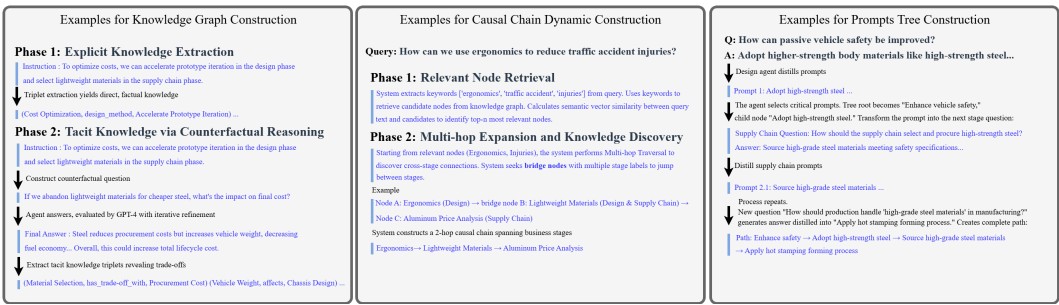

Figure 9: Examples of implementing a core component in the Cochain framework.

Q: To improve fuel economy, the design team recommends that the new vehicle model adopt carbon fiber materials to replace steel. What is your recommendation?

**Examples of Single Agent**

A: Carbon fiber is an ideal choice. It is 60% lighter than steel, 5 times stronger, and has excellent corrosion resistance, which can significantly improve fuel economy and handling. In addition, carbon fiber has a highly technological appearance and is an iconic material for luxury models. Therefore, it is recommended to adopt it comprehensively to maximize performance advantages.

**Examples of Complete Workflow of Cochain**

Cochain's working mechanism: first retrieves knowledge graphs and performs causal chain exploration, then prompts tree retrieval.

**Knowledge Graphs Retrieval**

(Lightweight, improves, fuel economy)
(Carbon fiber, property, lightweight)
(Carbon fiber, property, high strength)
(Carbon fiber, requires, complex molding process)
(Carbon fiber, faces, scarce suppliers)
...

**Causal Chain**

Carbon fiber requires complex molding process, complex molding process relates to redesigning connection structures; Complex molding process requires new curing equipment, new curing equipment considers compatibility with stamping lines ...

**Prompts Tree**

User Need: New material application → Design: Analyze design impacts and challenges → Supply Chain: Assess supply chain feasibility → Production: Consider existing production processes → QI: Establish quality standards for new materials; User Need: Improve fuel economy → Design: Lightweight body design → Supply Chain: Assess supply chain feasibility → Production: Improve production processes to ensure processing quality → QI: Inspect product quality after production process improvements ...

**Answer by Cochain**

We recommend adopting a phased strategy, first validating carbon fiber processes on non-structural components such as spoilers and interior panels, and then expanding after costs become controllable. In design, although lightweight effects are good, **the molding process is complex, requiring redesign of connection structures and crash energy absorption zones**; the supply chain **faces challenges of scarce suppliers, prices far exceeding steel, and long procurement cycles**; production requires **investment in new curing equipment and consideration of compatibility with existing stamping lines**; quality control requires **addition of new inspection items such as fiber direction and curing degree**.

Figure 10: Examples of single agent and complete workflow of Cochain

# B    PROOF: TACIT KNOWLEDGE & COUNTERFACTUAL OUTPUT MODEL

Given the inherent difficulty in directly extracting hidden layer knowledge from agents operating within complex systems, our approach introduces counterfactual reasoning as a means to probe and subsequently infer tacit knowledge. This section elucidates the probabilistic formulations developed to model these inferred internal aspects of each agent. It details the conceptualization of tacit knowledge, the generation of counterfactual outputs by the agent, and the overall generative process used to understand these unobservable agent workings.

The foundation of our approach to modeling tacit knowledge within each agent lies in its conditional generation based on latent tacit variables. We define $h_{ij}$ as the tacit knowledge specific to a sample $(X_{ij}, Y_{ij})$ processed by a particular agent. Within an agent, $h_{ij}$ represents an internal state or representation that is not directly accessible. While $h_{ij}$ is an abstract construct for modeling these unobservable aspects, for illustrative purposes, one might conceptualize it as akin to intermediate computational states, learned feature representations, or internal configurations within the Agent that influence its output generation. Concurrently, $\theta_{ij}$ is defined as a latent tacit variable associated with the same sample and agent. This variable $\theta_{ij}$ is also an abstract construct, encapsulating underlying factors hypothesized to influence $h_{ij}$, such as specific input characteristics processed by the Agent, its operational modes, or conditioning contexts it operates under. The introduction of $\theta_{ij}$ and $h_{ij}$ is an attempt to create a probabilistic model for the Agent's internal, unobservable precursors to decision-making or generation.

A core assumption of our model is that this tacit knowledge $h_{ij}$ is not deterministically derived but is rather a stochastic realization conditioned on $\theta_{ij}$. This is particularly relevant for complex Agents, where output generation can be influenced by internal stochastic mechanisms or intricate decision processes. This probabilistic relationship is formally expressed as:

$$h_{ij} \mid \theta_{ij} \sim P(h_{ij} \mid \theta_{ij}) \tag{8}$$

This notation signifies that $h_{ij}$ is a random variable whose probability distribution $P(h_{ij} \mid \theta_{ij})$ is conditional upon the specific value or state of $\theta_{ij}$. The nature of this distribution would be specific to the agent and reflects the inherent complexities and uncertainties in how its internal states are formed. For instance, $\theta_{ij}$ might represent an abstract control signal or a high-level interpretation of input patterns by the agent, and $P(h_{ij} \mid \theta_{ij})$ would describe the distribution of possible internal representations resulting from it. This formulation acknowledges that even with a defined influencing factor $\theta_{ij}$, the precise tacit knowledge $h_{ij}$ (the agent's internal processing focus or state) can vary. The characteristics of $P(h_{ij} \mid \theta_{ij})$ can be explored and potentially learned through systematic counterfactual probing of the agent.

Following the generation of tacit knowledge, we model the generation of the counterfactual output $\tilde{Y}_{ij}$ by the agent. This output is conditioned not only on the counterfactual input $\tilde{X}_{ij}$ but also critically on the realized tacit knowledge $h_{ij}$ derived from Equation 8. We define $\tilde{Y}_{ij}$ as the Agent's generated output when presented with the counterfactual input $\tilde{X}_{ij}$, and $h_{ij}$ acts as the mediating internal state through which $\tilde{X}_{ij}$ is processed to produce $\tilde{Y}_{ij}$.

The generation of $\tilde{Y}_{ij}$ by such agents is often an inherently probabilistic process. Even with a given counterfactual input $\tilde{X}_{ij}$ and a specific internal tacit knowledge state $h_{ij}$, the agent may not produce a single, deterministic output. This inherent stochasticity is captured by the following conditional probability distribution, whose parameters might also be inferred from counterfactual observations:

$$\tilde{Y}_{ij} \mid h_{ij}, \tilde{X}_{ij} \sim P(\tilde{Y}_{ij} \mid h_{ij}, \tilde{X}_{ij}) \tag{9}$$

This expression states that $\tilde{Y}_{ij}$ (the counterfactual output) is a random variable whose distribution $P(\tilde{Y}_{ij} \mid h_{ij}, \tilde{X}_{ij})$ depends on both the specific tacit knowledge state $h_{ij}$ and the counterfactual input $\tilde{X}_{ij}$. The tacit knowledge $h_{ij}$ here serves as a crucial intermediary. For agents producing sequential outputs, $P(\tilde{Y}_{ij} \mid h_{ij}, \tilde{X}_{ij})$ might be represented as an autoregressive factorization, $P(\tilde{Y}_{ij} \mid h_{ij}, \tilde{X}_{ij}) = \prod_t P(\tilde{y}_{ij,t} \mid \tilde{y}_{ij,<t}, h_{ij}, \tilde{X}_{ij})$.

To determine the overall probability of observing a counterfactual output $\tilde{Y}_{ij}$ from an agent given only the counterfactual input $\tilde{X}_{ij}$—a common scenario when internal states are not directly observable—we must account for the entire hypothesized generative pathway. This involves marginalizing

out the unobserved latent tacit variable $\theta_{ij}$ and, by extension, the intermediate tacit knowledge $h_{ij}$. The final generation process is thus expressed as:

$$P(\tilde{Y}_{ij} \mid \tilde{X}_{ij}) = \int_{\Theta} P(\tilde{Y}_{ij} \mid h_{ij}, \tilde{X}_{ij})P(h_{ij} \mid \theta_{ij})P(\theta_{ij} \mid \tilde{X}_{ij})d\theta_{ij} \qquad (10)$$

This equation can be derived by considering the dependencies established in Equations 8 and 9.

The derivation proceeds as follows: First, we apply the law of total probability to marginalize over the latent tacit variable $\theta_{ij}$. The probability of $\tilde{Y}_{ij}$ given $\tilde{X}_{ij}$ can be written as an integral over all possible values of $\theta_{ij}$ in its domain $\Theta$:

$$P(\tilde{Y}_{ij} \mid \tilde{X}_{ij}) = \int_{\Theta} P(\tilde{Y}_{ij}, \theta_{ij} \mid \tilde{X}_{ij})d\theta_{ij}$$

Using the definition of conditional probability, $P(A, B \mid C) = P(A \mid B, C)P(B \mid C)$, we can rewrite the integrand $P(\tilde{Y}_{ij}, \theta_{ij} \mid \tilde{X}_{ij})$ as:

$$P(\tilde{Y}_{ij}, \theta_{ij} \mid \tilde{X}_{ij}) = P(\tilde{Y}_{ij} \mid \theta_{ij}, \tilde{X}_{ij})P(\theta_{ij} \mid \tilde{X}_{ij})$$

Here, $P(\theta_{ij} \mid \tilde{X}_{ij})$ represents the prior probability distribution of the tacit variable $\theta_{ij}$ given the counterfactual input $\tilde{X}_{ij}$. This term reflects how a given counterfactual input might stochastically lead to different internal configurations or "operational modes" $\theta_{ij}$ within the Agent that are not directly observable. The nature of this prior can also be investigated through counterfactual analysis. Substituting this back, we get:

$$P(\tilde{Y}_{ij} \mid \tilde{X}_{ij}) = \int_{\Theta} P(\tilde{Y}_{ij} \mid \theta_{ij}, \tilde{X}_{ij})P(\theta_{ij} \mid \tilde{X}_{ij})d\theta_{ij} \qquad (11)$$

The term $P(\tilde{Y}_{ij} \mid \theta_{ij}, \tilde{X}_{ij})$ is the probability of generating $\tilde{Y}_{ij}$ given both $\theta_{ij}$ and $\tilde{X}_{ij}$. Our model posits a specific structure for this, reflecting a Markov chain-like dependency: $\theta_{ij} \to h_{ij} \to \tilde{Y}_{ij}$ (all conditioned on $\tilde{X}_{ij}$ where appropriate). This structure implies two key conditional independence assumptions:

1. Given $\theta_{ij}$, $h_{ij}$ is independent of $\tilde{X}_{ij}$ if all influence of $\tilde{X}_{ij}$ on $h_{ij}$ is mediated through $\theta_{ij}$. More commonly, $P(h_{ij} \mid \theta_{ij})$ is defined as the direct influence of $\theta_{ij}$ on $h_{ij}$.

2. Given $h_{ij}$ and $\tilde{X}_{ij}$, $\tilde{Y}_{ij}$ is independent of $\theta_{ij}$. That is, $P(\tilde{Y}_{ij} \mid h_{ij}, \theta_{ij}, \tilde{X}_{ij}) = P(\tilde{Y}_{ij} \mid h_{ij}, \tilde{X}_{ij})$. The tacit knowledge $h_{ij}$ fully mediates the influence of $\theta_{ij}$ on $\tilde{Y}_{ij}$.

Under these assumptions, we can decompose $P(\tilde{Y}_{ij} \mid \theta_{ij}, \tilde{X}_{ij})$:

$$P(\tilde{Y}_{ij} \mid \theta_{ij}, \tilde{X}_{ij}) = \int P(\tilde{Y}_{ij}, h_{ij} \mid \theta_{ij}, \tilde{X}_{ij})dh_{ij}$$

$$= \int P(\tilde{Y}_{ij} \mid h_{ij}, \theta_{ij}, \tilde{X}_{ij})P(h_{ij} \mid \theta_{ij}, \tilde{X}_{ij})dh_{ij}$$

Applying the conditional independence assumptions:

$$= \int P(\tilde{Y}_{ij} \mid h_{ij}, \tilde{X}_{ij})P(h_{ij} \mid \theta_{ij})dh_{ij}$$

The form in Equation 10 implies that for a given $\theta_{ij}$ in the outer integral, the relevant $h_{ij}$ is the one generated from that $\theta_{ij}$, effectively collapsing the inner integral. Thus, $P(\tilde{Y}_{ij} \mid \theta_{ij}, \tilde{X}_{ij})$ is represented by the product $P(\tilde{Y}_{ij} \mid h_{ij}, \tilde{X}_{ij})P(h_{ij} \mid \theta_{ij})$, where $h_{ij}$ is the specific realization tied to $\theta_{ij}$ within the integral. Substituting this construction into Equation 11 directly yields Equation 10:

$$P(\tilde{Y}_{ij} \mid \tilde{X}_{ij}) = \int_{\Theta} P(\tilde{Y}_{ij} \mid h_{ij}, \tilde{X}_{ij})P(h_{ij} \mid \theta_{ij})P(\theta_{ij} \mid \tilde{X}_{ij})d\theta_{ij}$$

This integral sums the probabilities of $\tilde{Y}_{ij}$ occurring via all possible latent tacit variables $\theta_{ij}$. Each path's contribution is weighted by the prior probability of $\theta_{ij}$, the probability of the corresponding tacit knowledge $h_{ij}$ arising from that $\theta_{ij}$, and finally the probability of the output $\tilde{Y}_{ij}$ given that $h_{ij}$ and the input $\tilde{X}_{ij}$.

## C  IMPLEMENTATION DETAILS

**Definition of Business Workflow:** It is a complex process with many interconnected stages. These stages are specialized. The key feature is that these stages strongly depend on each other. A decision in one stage requires more than just expert knowledge of that area. It must also anticipate and use knowledge and constraints from the other stages.

Table 7 presents a comprehensive comparison of Cochain against other baseline models. Cochain demonstrates superior agent collaboration, characterized by faster inference speeds and lower hallucination rates. More critically, Cochain exhibits decomposability, ensuring that the collaboration process is resilient and does not terminate due to failures in individual components. It is important to note that for CoA, source documents are not included due to the differing task orientation.

The LLMs referenced in Figure 7 are fine-tuned using QLoRA (Dettmers et al., 2023), configured with a LoRA rank of 8 and trained for 3 epochs, on two H100 GPUs, with models quantized to 4-bit precision via bitsandbytes. Other open-source models are fine-tuned for 10 epochs using LoRA (Hu et al., 2022). The inference times reported in Figure 5 and Table 8 are benchmarked by deploying the open-source models on a single A40 GPU to mitigate network variability.

Table 7: Comparison of different multi-agent collaboration methods

| Baseline | Collaboration Method | Decomposable | Reasoning Speed | Hallucination Rate | Agent |
|---|---|---|---|---|---|
| PMC (Zhang et al., 2025a) | Hierarchical Planning | ✗ | Low | Middle | Multiple |
| MedAgents (Tang et al., 2024) | Discussion & Voting | ✗ | Low | Middle | Multiple |
| Debate(short) (Du et al., 2024) | Debate | ✗ | Low | Middle | Multiple |
| Debate(long) (Du et al., 2024) | Debate | ✗ | Low | Middle | Multiple |
| CoA (Zhang et al., 2024c) | Chain Collaboration | ✗ | Low | Middle | Multiple |
| Original | - | ✓ | High | High | Single |
| Cochain | Knowledge Fusion | ✓ | High | Low | Multiple |

The appendix provides a more extensive report on experimental results across additional metrics. These include BERTScore-P, BERTScore-R, BLEU-4 (Papineni et al., 2002) for measuring text alignment, METEOR (Banerjee & Lavie, 2005) which considers semantically equivalent phrases, ROUGE-2 for assessing content breadth, and ROUGE-L (Lin, 2004) which evaluates deep semantic alignment, particularly for long texts.

## D  MORE EXPERIMENTS AND ANALYSIS

### D.1  MORE EFFICIENCY AND COST ANALYSIS

Table 8 presents a comparative analysis of Cochain against other multi-agent systems, focusing on key operational metrics: Inference Time, Token Throughput, and Cost. A noteworthy observation on the Automotive and E-commerce datasets is that Cochain's average input token count is approximately one-twentieth of that utilized by PMC and CoA. This disparity suggests that these alternative systems expend a considerable volume of tokens on inter-agent communication. In contrast, Cochain innovatively transforms such communication into structured knowledge graphs and prompts tree, a mechanism that effectively curtails operational costs. The overall results compellingly demonstrate that Cochain achieves substantial reductions in inference time, token throughput, and aggregate cost when benchmarked against other multi-agent approaches. **Remarkably, the efficiency metrics for Cochain closely approximate those observed for single-agent systems, and its average cost is even lower**, underscoring a significant advantage of our proposed framework in terms of computational resource utilization and operational expenditure. Table 9 presents the retrieval contribution of each component during the inference process. The prompt tree retrieval employs a depth-first search (DFS) strategy. In terms of time complexity, this is equivalent to a simple path traversal with a complexity of $O(D)$, where $D$ denotes the depth of the business process.

Table 8: Comparative analysis of inference time, token throughput, and cost. [†]Inference time is measured using a locally deployed Qwen2-7B model to obviate network latency associated with API calls. Conversely, token throughput and cost metrics are derived using Claude-3.5-Haiku.

| Baselines | [†]Inference Time (s) | Avg.Input | Avg.Output | Avg.Cost ($) |
|---|---|---|---|---|
| **Automotive** | | | | |
| PMC (Zhang et al., 2025a) | 241.56 | 11637.94 | 6644.22 | 0.0359 |
| MedAgents (Tang et al., 2024) | 196.53 | 3901.03 | 3483.24 | 0.0170 |
| Debate(short) (Du et al., 2024) | 187.16 | 3938.35 | 4809.27 | 0.0223 |
| Debate(long) (Du et al., 2024) | 162.57 | 4357.87 | 5278.53 | 0.0246 |
| CoA (Zhang et al., 2024c) | 227.21 | 11428.06 | 5446.62 | 0.0309 |
| Original | 37.22 | 93.49 | 1041.38 | 0.0042 |
| Cochain | 38.77 | 501.62 | 541.74 | **0.0026** |
| **Pharmaceutical** | | | | |
| PMC (Zhang et al., 2025a) | 198.49 | 3796.07 | 2613.73 | 0.0135 |
| MedAgents (Tang et al., 2024) | 169.53 | 3504.85 | 3504.85 | 0.0168 |
| Debate(short) (Du et al., 2024) | 548.66 | 3372.56 | 3772.79 | 0.0178 |
| Debate(long) (Du et al., 2024) | 574.95 | 3810.87 | 4268.74 | 0.0201 |
| CoA (Zhang et al., 2024c) | 253.18 | 5493.94 | 3450.45 | 0.0182 |
| Original | 60.96 | 77.28 | 1091.32 | 0.0044 |
| Cochain | 44.70 | 533.88 | 830.34 | **0.0037** |
| **E-commerce** | | | | |
| PMC (Zhang et al., 2025a) | 256.84 | 11319.74 | 6452.89 | 0.0349 |
| MedAgents (Tang et al., 2024) | 181.61 | 3816.26 | 3445.34 | 0.0169 |
| Debate(short) (Du et al., 2024) | 318.51 | 4414.23 | 5660.94 | 0.0261 |
| Debate(long) (Du et al., 2024) | 293.42 | 5438.52 | 6911.85 | 0.0321 |
| CoA (Zhang et al., 2024c) | 195.49 | 11446.05 | 5353.24 | 0.0306 |
| Original | 35.83 | 60.90 | 1080.92 | 0.0044 |
| Cochain | 37.10 | 561.80 | 620.15 | **0.0029** |

Table 9: Latency Breakdown Analysis of Cochain's Inference Components

| Inference Component | Average Time (s) | Percentage of Total Time (%) |
|---|---|---|
| Knowledge Graph Retrieval | 0.6200 s | 1.57 % |
| Prompts Tree Retrieval | 0.0022 s | <0.01 % |
| LLM Model Inference | 38.7700 s | 98.42 % |

## D.2 MORE EXPERIMENTS ON SINGLE AGENT

Table 10 and Table 11 demonstrate that integrating Cochain markedly enhances single-agent performance across diverse backbone models during two distinct stages of a business workflow. This enhancement is consistently reflected across all evaluated metrics, irrespective of whether the LLMs are in their raw, pre-trained state or have undergone fine-tuning. Notably, a synergistic effect between fine-tuning and the application of Cochain typically yields optimal performance, underscoring Cochain's robust capability to leverage specialized model knowledge. The broad-spectrum improvements observed across various model metrics highlight the effectiveness and versatility of Cochain in mitigating prior issues of undercollaboration.

Table 10: The performance of our proposed method on single-agent. BERTScore F1 is abbreviated as BS-F, BERTScore Precision as BS-P, and BERTScore Recall as BS-R. We highlight the best and second-best results.

| Backbone | Method | BS-F | BS-P | BS-R | BLEU-4 | GLEU | METEOR | ROUGE-2 | ROUGE-L |
|---|---|---|---|---|---|---|---|---|---|
| Llama2-7B | Raw LLM | 62.25 | 64.41 | 60.54 | 6.04 | 8.28 | 17.52 | 7.78 | 15.64 |
| | + Cochain | 65.70 | 68.25 | 63.50 | 10.99 | 13.84 | 25.50 | 10.44 | 16.46 |
| | Finetuned LLM | 70.68 | 70.71 | 70.67 | 14.31 | 18.79 | 35.03 | 11.97 | 21.49 |
| | + Cochain | 71.46 | 71.35 | 71.59 | 16.50 | 20.96 | 38.69 | 13.79 | 21.58 |
| Qwen2-7B | Raw LLM | 71.76 | 71.79 | 71.75 | 17.23 | 22.30 | 41.13 | 12.88 | 24.00 |
| | + Cochain | 75.24 | 74.90 | 75.58 | 22.32 | 26.51 | 52.00 | 19.53 | 27.94 |
| | Finetuned LLM | 71.72 | 71.75 | 71.70 | 16.90 | 21.64 | 40.77 | 13.35 | 24.07 |
| | + Cochain | 75.39 | 75.14 | 75.65 | 24.17 | 28.17 | 51.22 | 21.22 | 28.87 |
| Llama3-8B | Raw LLM | 68.93 | 68.38 | 69.59 | 7.47 | 9.14 | 24.48 | 10.99 | 19.28 |
| | + Cochain | 73.75 | 73.36 | 74.17 | 19.47 | 23.52 | 48.43 | 17.57 | 25.10 |
| | Finetuned LLM | 70.89 | 70.88 | 70.91 | 14.90 | 19.05 | 39.83 | 12.59 | 22.90 |
| | + Cochain | 74.56 | 74.13 | 75.01 | 22.12 | 26.15 | 50.59 | 19.41 | 27.19 |
| Llama2-13B | Raw LLM | 60.79 | 63.16 | 59.00 | 5.73 | 7.96 | 16.90 | 7.52 | 15.43 |
| | + Cochain | 67.47 | 70.20 | 65.15 | 13.42 | 16.88 | 29.97 | 13.65 | 18.57 |
| | Finetuned LLM | 70.88 | 70.88 | 70.90 | 14.57 | 19.14 | 35.87 | 12.19 | 21.85 |
| | + Cochain | 71.80 | 72.08 | 71.58 | 18.16 | 22.29 | 40.33 | 14.96 | 23.18 |
| Qwen2.5-14B | Raw LLM | 71.86 | 72.03 | 71.70 | 9.48 | 12.16 | 40.71 | 11.85 | 17.86 |
| | + Cochain | 75.62 | 75.56 | 75.69 | 12.39 | 14.71 | 47.95 | 17.66 | 19.55 |
| | Finetuned LLM | 71.50 | 71.48 | 71.53 | 9.25 | 11.75 | 40.16 | 12.01 | 17.67 |
| | + Cochain | 75.18 | 75.00 | 75.38 | 13.11 | 15.08 | 48.35 | 19.70 | 20.30 |

Table 11: The performance of our proposed method. The results of this table are in a different stage, as compared to the stage presented in Table 10. We highlight the best and second-best results.

| Backbone | Method | BS-F | BS-P | BS-R | BLEU-4 | GLEU | METEOR | ROUGE-2 | ROUGE-L |
|---|---|---|---|---|---|---|---|---|---|
| Llama2-7B | Raw LLM | 61.76 | 64.72 | 59.44 | 6.36 | 8.69 | 18.25 | 8.66 | 16.17 |
| | + Cochain | 63.57 | 66.31 | 61.23 | 9.49 | 12.37 | 23.12 | 8.99 | 17.46 |
| | Finetuned LLM | 69.88 | 69.78 | 70.00 | 12.02 | 17.02 | 32.59 | 9.26 | 20.69 |
| | + Cochain | 70.47 | 70.43 | 70.55 | 13.79 | 18.58 | 35.44 | 10.91 | 20.26 |
| Qwen2-7B | Raw LLM | 70.34 | 71.12 | 70.60 | 13.74 | 19.88 | 35.29 | 9.47 | 22.01 |
| | + Cochain | 73.48 | 72.37 | 74.63 | 15.77 | 19.21 | 50.38 | 16.25 | 21.99 |
| | Finetuned LLM | 70.28 | 70.08 | 70.50 | 13.61 | 18.95 | 35.07 | 9.96 | 22.38 |
| | + Cochain | 74.51 | 74.30 | 74.74 | 21.99 | 25.76 | 50.49 | 20.71 | 26.00 |

As shown in Table 12, to better quantify the impact of insufficient collaboration, we compare the baseline single agent to one provided with minimal communication, simulated by manually injecting five prompts containing cross-domain constraints. The results of this table clearly show: First, under-collaboration is a real problem, as the agent with injected communication showed a slight performance improvement over the original agent. This directly proves that adding cross-domain communication improves outcomes, thereby quantifying the performance loss from under-collaboration. Second, Cochain's intelligent collaboration is far superior to simple communication, as the performance improved substantially after applying Cochain. This demonstrates that Cochain's value is not just about adding information. Rather, it excels at intelligently filtering, organizing, and structuring collaborative knowledge. Its effectiveness far surpasses that of unstructured, manually added information and helps to avoid the risk of over-collaboration.

Table 12: Quantifying under collaboration.

| Backbone | Method | BS-F | GLEU | ROUGE-L |
|---|---|---|---|---|
| Qwen2-7B | Single Agent | 71.72 | 21.64 | 24.07 |
| | + Manual Prompts | 72.54 (↑) | 21.83 (↑) | 24.34 (↑) |
| | **+ Cochain** | **75.39 (⇑)** | **28.17 (⇑)** | **28.87 (⇑)** |
| DeepSeek-R1-7B | Single Agent | 71.35 | 19.55 | 22.71 |
| | + Manual Prompts | 71.83 (↑) | 19.90 (↑) | 23.38 (↑) |
| | **+ Cochain** | **73.19 (⇑)** | **21.71 (⇑)** | **24.01 (⇑)** |

### D.3 MORE EXPERIMENTS ON SPECIALIZED AGENT COLLABORATION

Table 5 presents the results from a distinct experimental configuration designed to further investigate the coordination capabilities of Cochain. Whereas Table 10 assessed a multi-domain model (a single Qwen2-7B model fine-tuned on all domains) and specialized models (different models each fine-tuned for specific domains), Table 13 exclusively focuses on a setup utilizing multiple agents of the same backbone, each independently fine-tuned for a specific domain. The findings indicate that Cochain effectively facilitates collaboration among these identically architected yet uniquely specialized agents. Performance enhancements are observed both within their respective specialized domains and in the quality of the comprehensive, integrated solutions. This underscores Cochain's distinct advantage in synergizing specialized knowledge from a team of uniquely trained agents that share a common architectural foundation.

Table 13: Results for one model at particular stages. This means that we use Qwen2-7B for four fine-tunings, and the four fine-tuned models are experimented with at their respective stages. We highlight the fine-tuned results, Cochain application, and improvements(↑).

| Backbone | Stage | Method | BS-F | BS-P | BS-R | BLEU-4 | GLEU | ROUGE-L |
|---|---|---|---|---|---|---|---|---|
| Qwen2-7B | $S_1$ | Finetuned | 71.72 | 71.75 | 71.70 | 16.90 | 21.64 | 24.07 |
| | | + Cochain | 75.39 | 75.14 | 75.65 | 24.17 | 28.17 | 28.87 |
| | | ↑ | +3.67 | +3.39 | +3.95 | +7.27 | +6.53 | +4.80 |
| | $S_2$ | Finetuned | 70.24 | 70.47 | 70.16 | 13.19 | 18.29 | 20.95 |
| | | + Cochain | 74.23 | 74.01 | 74.46 | 20.05 | 23.57 | 24.71 |
| | | ↑ | +3.99 | +3.54 | +4.30 | +6.86 | +5.28 | +3.76 |
| | $S_3$ | Finetuned | 70.28 | 70.08 | 70.50 | 13.61 | 18.95 | 22.38 |
| | | + Cochain | 74.51 | 74.30 | 74.74 | 21.99 | 25.76 | 26.00 |
| | | ↑ | +4.23 | +4.22 | +4.24 | +8.38 | +6.81 | +3.62 |
| | $S_4$ | Finetuned | 69.40 | 69.49 | 69.32 | 13.51 | 18.26 | 21.82 |
| | | + Cochain | 75.06 | 74.99 | 75.15 | 23.95 | 27.70 | 27.49 |
| | | ↑ | +5.66 | +5.50 | +5.83 | +10.44 | +9.44 | +5.67 |
| | Score | | 6.4 | | | 9.5 | | +3.1 |

### D.4 MORE RESULTS FROM ABLATION STUDIES

On the automotive dataset, we report additional ablation study results focusing on the impact of different backbone models. As indicated in Table 14, "w/o Prompts Tree" markedly impaired performance. Specifically, when applied to the Llama-8B backbone, this ablation led to a 1.97% reduction in the BS-F score, with notable degradations also observed across other evaluation metrics. These findings are consistent with the experimental results presented in Table 6, further underscoring the critical role of the Prompts Tree component in our framework.

Table 14: More results of ablation study. $\mathcal{KG}$ represents the knowledge graph. We highlight the most and second-most efficient modules.

| Backbone | Method | BS-F | BS-P | BS-R | BLEU-4 | GLEU | METEOR | ROUGE-2 | ROUGE-L |
|---|---|---|---|---|---|---|---|---|---|
| Llama2-7B | Cochain | 71.46 | 71.35 | 71.59 | 16.50 | 20.96 | 38.69 | 13.79 | 21.58 |
| | w/o $\mathcal{KG}$ | 71.26 | 71.14 | 71.41 | 16.06 | 20.53 | 38.21 | 13.55 | 21.50 |
| | w/o Causal Chain | 71.26 | 71.13 | 71.44 | 15.63 | 19.59 | 40.32 | 14.33 | 21.16 |
| | w/o Prompts Tree | 71.06 | 70.92 | 71.20 | 15.20 | 19.61 | 36.66 | 13.07 | 21.38 |
| Qwen2-7B | Cochain | 75.39 | 75.14 | 75.65 | 24.17 | 28.17 | 51.22 | 21.22 | 28.87 |
| | w/o $\mathcal{KG}$ | 75.21 | 74.91 | 75.53 | 23.56 | 27.81 | 49.76 | 20.15 | 28.74 |
| | w/o Causal Chain | 75.20 | 74.92 | 75.50 | 23.68 | 27.55 | 51.00 | 21.14 | 28.42 |
| | w/o Prompts Tree | 73.08 | 73.20 | 72.98 | 18.40 | 22.45 | 45.43 | 16.06 | 24.56 |
| Llama3-8B | Cochain | 74.56 | 74.13 | 75.01 | 22.12 | 26.15 | 50.59 | 19.41 | 27.19 |
| | w/o $\mathcal{KG}$ | 74.27 | 73.80 | 74.75 | 21.45 | 25.46 | 49.99 | 19.09 | 27.13 |
| | w/o Causal Chain | 73.63 | 72.79 | 74.50 | 17.34 | 20.70 | 49.15 | 18.28 | 23.78 |
| | w/o Prompts Tree | 72.59 | 72.45 | 72.75 | 17.01 | 21.02 | 45.00 | 15.18 | 23.83 |
| Qwen2.5-14B | Cochain | 75.18 | 75.00 | 75.38 | 13.11 | 15.08 | 48.35 | 19.70 | 20.30 |
| | w/o $\mathcal{KG}$ | 74.83 | 74.41 | 75.25 | 12.49 | 14.55 | 47.32 | 18.39 | 19.76 |
| | w/o Causal Chain | 74.71 | 74.66 | 74.79 | 12.28 | 14.17 | 47.65 | 19.30 | 19.44 |
| | w/o Prompts Tree | 73.01 | 73.01 | 73.02 | 9.98 | 12.25 | 42.72 | 14.15 | 18.00 |

## D.5 Integration of Cochain with Baselines

We further investigate the integration of Cochain with existing baseline methods, specifically PMC and MedAgents, through experiments conducted on a pharmaceutical dataset using Claude-3.5-Haiku. As reported in Table 15, the combination of Cochain with PMC results in improvements of 2.62% and 7.55% in the BS-F and BLEU-4 scores, respectively. This outcome further substantiates the efficacy of Cochain in mitigating issues associated with over-collaboration.

Table 15: The improvement of Cochain over other baselines on pharmaceutical datasets.

| Baseline | BS-F | BS-P | BS-R | BLEU-4 | GLEU | METEOR | ROUGE-2 | ROUGE-L |
|---|---|---|---|---|---|---|---|---|
| PMC (Zhang et al., 2025a) | 66.51 | 66.06 | 66.97 | 5.17 | 12.33 | 20.31 | 5.48 | 13.56 |
| PMC (Zhang et al., 2025a) + Cochain | **69.13** | **68.49** | **69.80** | **12.72** | **18.48** | 27.01 | **6.93** | **15.82** |
| MedAgents (Tang et al., 2024) | 66.23 | 66.02 | 66.48 | 7.17 | 14.02 | 22.59 | 5.43 | 14.32 |
| MedAgents (Tang et al., 2024) + Cochain | 67.45 | 66.91 | 68.00 | 11.65 | 17.80 | **27.06** | 6.16 | 15.74 |

## D.6 Evaluation of Collaborative Knowledge Graph

To evaluate the reliability of our final knowledge graph, we randomly sampled 2,000 triplets from it, consisting of 1,000 from explicit knowledge extraction and 1,000 from our counterfactual method. We used three core metrics for evaluation: Factual Correctness (Is the knowledge factually correct in the real world?),Task Relevance (How helpful is the knowledge for solving business workflow problems?), and Knowledge Depth (Is the knowledge common sense, or does it require professional insight?) We invited two domain experts for a blind review. We also used Gemini-2.5-Pro as a judge for an objective evaluation. The average scores (out of 5) are presented in the Table 16.

Table 16: Evaluation of Collaborative Knowledge Graphs by Human Experts and LLMs.

| Evaluation Metric | Human Expert Score (Avg. / 5.0) | Gemini-2.5-Pro Score (Avg. / 5.0) |
|---|---|---|
| Factual Correctness | 4.99 | 5.00 |
| Task Relevance | 4.76 | 4.81 |
| Knowledge Depth | 4.31 | 4.54 |

## D.7 Overcollaboration is More Serious than Undercollaboration

Business workflow tasks are particularly susceptible to the "overcollaboration" phenomenon. When multiple agents collaborate, this manifests as responses deviating from core issues and reducing

answer quality. Comparative analysis of Table 1, Table 5, and additional single-agent experimental results (Appendix D.2) demonstrates that, despite consuming significantly more computational resources, overcollaboration performs substantially worse than undercollaboration. Overcollaboration's challenging nature is its covertness—systems maintain high activity and apparent collaboration, making the problem hard to detect and rectify promptly. While task decomposition, such as approaches like PMC, and summarization, such as strategies like CoA, are considered effective methods for focusing on core issues (Wu et al., 2024; Chen et al., 2024b), Cochain's collaboration mechanism exhibits superior performance in controlling excessive collaboration, enabling a more precise focus on critical task requirements.

### D.8 ROBUSTNESS TO INCOMPLETE RETRIEVAL

To validate the system's robustness against potential keyword retrieval failures, we conducted a stress test. We designed 50 structured queries in the format of "How to use [Core Concept A] to solve problems related to [Target Concept B]". In the test condition, we deliberately masked all keywords corresponding to [Target Concept B] to simulate a first-stage retrieval failure. For instance, in the query "How can ergonomics be used to reduce traffic accident injuries?", the keywords "traffic" and "accident" were blocked.

While end-to-end semantic retrieval could theoretically solve keyword-mismatch issues, it becomes computationally prohibitive for real-time interaction on massive knowledge graphs. We therefore adopt the Retrieval-Ranking two-stage paradigm, a proven industry standard (Covington et al., 2016). This approach combines the advantages of both methods: keyword-based retrieval in the first stage ensures precise matching of technical terms and proper nouns, avoiding omissions from semantic over-generalization. In the second stage, semantic ranking captures the query's overall intent, ensuring system generalization and robustness.

Table 17: Performance comparison under simulated retrieval failure. Masking the target concept's keywords results in a negligible performance drop, demonstrating the system's robustness.

| Test Condition | BS-F | GLEU | ROUGE-L |
|---|---|---|---|
| Full Query | 74.16 | 22.08 | 25.86 |
| Target Concept Masked | 73.87 | 21.42 | 24.57 |
| **Performance Drop (%)** | **-0.29%** | **-0.66%** | **-1.29%** |

The results in Table 17 confirm the effectiveness of this design. Even when forcing a first-stage failure, the BS-F score experienced a minimal drop of only 0.29%, from 74.16 to 73.87, with similarly small decreases in GLEU (-0.66%) and ROUGE-L (-1.29%) scores. This resilience is attributed to two factors. First, as justified above, the semantic ranking stage corrects for an imperfect initial candidate set. More critically, the subsequent graph traversal and causal chain stages provide a powerful secondary path to completeness. Even if "Traffic Accident" is missed during retrieval, it can be discovered through pre-existing knowledge paths represented as triples (e.g., (Injuries, is_a_result_of, Traffic Accident)), ensuring the final context remains robust and comprehensive.

## E CASE STUDY

### E.1 INTERPRETABILITY ANALYSIS

We investigate the interpretability characteristics of the model after the application of Cochain through a case study. As shown in Table 18, we employ the same color to annotate logically related content visually. The findings reveal that, following the application of Cochain, the model is capable of performing multi-dimensional reasoning based on knowledge graphs, causal inference chains, and contextual cues when addressing user needs, thereby generating outputs that are traceable, reliable, and interpretable. The application of the Cochain method has significantly enhanced the transparency and explainability of the model's reasoning process, providing empirical evidence for understanding the model's collaborative decision-making mechanisms.

### E.2 COMPARATIVE ANALYSIS OF MODEL OUTPUTS

To understand how different baselines handle complex instructions, as shown in Figure 11, we qualitatively analyzed their outputs against a reference answer, focusing on their integration of User Experience(UX) design within the automotive business workflow. Additionally, we present case studies for the pharmaceutical business workflow and the e-commerce business workflow, as shown in Figure 12 and Figure 13.

**Cochain: Clear Structure, Focused on UX Integration Process.** Cochain's response was notably well-structured, using a seven-point list for key UX integration stages, enhancing readability and providing a clear action framework. It comprehensively covered the UX lifecycle—from user research and design principles to interdisciplinary collaboration, technological innovation, continuous iteration, and performance-cost/quality assurance. Its explicit "collaborative perspective of the automotive business workflow" aligned with the reference answer's holistic view, effectively capturing its spirit by offering a structured, UX-centric methodology.

**Comparison of Cochain with Other Model Responses.** Cochain's primary strength is its sustained focus on the UX design integration process and principles, closely matching the prompt's core intent and the reference answer. In contrast, PMC and CoA lean towards high-level strategic planning (e.g., PLM platforms, KPIs), emphasizing system management rather than the deep UX integration across workflow stages highlighted by the reference answer. Debate showcases significant depth in UX concepts and cutting-edge technologies. However, its primary focus is on HMI innovation details, rather than systematically integrating UX across broader business workflow processes. Cochain can better balance innovation with this workflow integration. MedAgents share some structural similarities with Cochain, but Cochain is more direct in articulating the end-to-end UX lifecycle, especially "Continuous Iteration and Optimization," and its business workflow perspective. While other models excel in specific areas (e.g., Debate's UX tech depth; PMC and CoA's strategic scope), Cochain demonstrates the closest overall alignment with the reference answer's advocated UX-integrated process throughout the business workflow and its pragmatic perspective, thus performing excellently in this qualitative assessment.

Table 18: Case study of interpretability. The three distinct modules, namely Knowledge, Causal Chain, and Prompts Tree, are each assigned one of the three different color schemes: red, yellow, and blue. The colors within the modules correspond to those in the output, and logically related content is annotated using the same color scheme.

| User need | How can cost optimization be achieved in automotive manufacturing? |
|---|---|
| Knowledge | Cost optimization points include technological innovation, quality control, and environmental and energy management. Design optimization, material selection, and supply chain network construction strategies are measures for cost optimization... |
| Cause chain | Stamping is primarily used for component forming and can be integrated with automated and intelligent production lines, intelligent monitoring and optimization systems, as well as smart mold maintenance... |
| Prompts tree | **Prompt 1:** User Need: Cost optimization → Design Method: Accelerating prototype iteration → Supply Chain Method: Selection of lightweight materials → Production Method: Research and analysis of lightweight material performance and cost → Quality Inspection Method: ensuring materials meet performance and cost requirements. 
 **Prompt 2:** User Need: Cost optimization → Design Method: Accelerating prototype iteration → Supply Chain Method: Selection of lightweight materials → Production Method: Design optimization to enhance material utilization → Quality Inspection Method: Testing the strength and stability of the optimized structure. 
 ... |
| Output | In automotive design, it is essential to consider prototype optimization and innovation in design methodologies. For instance, to address customer-specific vehicle customization, cost control during the design phase can be achieved by accelerating prototype iterations and utilizing lightweight materials. Additionally, the properties and costs of lightweight materials should be taken into account to optimize structural design and enhance material utilization. 
 In supply chain management, cost optimization can also be achieved. This includes strategies for building the supply chain network and optimizing logistics. By consolidating procurement and leveraging economies of scale, procurement costs can be reduced. Collaborative Planning, Forecasting, and Replenishment (CPFR) methods can further optimize inventory costs. 
 In the manufacturing stage, cost optimization can be achieved through the introduction and implementation of automated and intelligent production lines. For example, the use of automated production lines, coupled with intelligent monitoring and optimization systems, can reduce redundant tasks and increase production efficiency. Furthermore, for specific manufacturing processes, such as stamping, material performance costs can be analyzed, and intelligent molds and automatic maintenance systems can be developed to optimize costs. 
 Finally, in the quality inspection stage, ensuring that all materials and products meet the predetermined performance and cost requirements is crucial. Additionally, improving production processes to ensure product processing quality is an important aspect of quality inspection. |

| Question | In the design process, how can User Experience (UX) design be more effectively integrated to improve the human-computer interaction quality of infotainment systems? |
|---|---|
| reference | In the automotive design process, to more effectively integrate User Experience (UX) design and enhance the human-computer interaction quality of infotainment systems, we need to consider the collaborative situation of the entire business workflow. We must first design with a user-centric approach, obtaining user needs and habits through methods such as user research, user interviews, questionnaires, behavioral observation, constructing user personas, and user journey maps. At the same time, understanding the user's environment and device habits is crucial for designing an automotive infotainment system that meets user needs and habits. During the design phase, we need to consider incorporating elements of User Experience (UX) design, including the integration of physical controls and digital interfaces, optimizing the infotainment system interface, and enhancing its intelligence and convenience. Simultaneously, we should draw on the latest technological trends to keep the infotainment system up-to-date. Regarding system performance, we also need to perform optimizations to extend its service life, enabling it to better meet users' long-term usage needs. Furthermore, we must also consider the selection of lightweight materials in the supply chain. This not only provides a better operational feel but also reduces the overall weight of the vehicle, further enhancing the user experience. In terms of production processes, we need to research and analyze the performance and cost of lightweight materials, optimize the design structure to improve material utilization, and refine production processes to ensure processing quality, thereby guaranteeing the quality and performance of the infotainment system. Finally, in the quality inspection phase, we need to ensure that the execution results of each stage meet the preset goals. This includes verifying whether materials meet performance and cost requirements, inspecting the strength and stability after structural optimization, and examining the product quality after improvements in production processes, ensuring that the infotainment system we provide can serve users stably and durably, enhancing their user experience. In summary, during the design process, we need to integrate multidisciplinary collaboration, including user research, design, supply chain management, production, and quality inspection, forming a collaborative work chain aimed at enhancing the user experience. |
| Cochain  BS-F:**73.75** BS-R:**74.06** BS-P:**73.44** GLEU:**24.27** METEOR:**34.39** BLEU-4:**20.27** ROUGE-2:**15.25** ROUGE-L:**24.69** | From a collaborative perspective based on the automotive business workflow, I will comprehensively answer how to effectively integrate User Experience (UX) design to enhance the human-computer interaction quality of infotainment systems from multiple dimensions. Firstly, in the User Research and Needs Analysis Phase, conduct comprehensive user research, including user interviews, questionnaires, and behavioral observation; construct detailed user personas and user journey maps; and deeply understand the driving habits and interaction preferences of different user groups. Secondly, regarding Design Principles and Methods, adopt a user-centric approach, focusing on the usability and intuitiveness of interaction design; ensure seamless integration of physical controls and digital interfaces; and balance technological innovation with actual user needs. Thirdly, for Interdisciplinary Collaboration, integrate professionals from multiple fields such as human-computer interaction, software engineering, and automotive design, and establish multi-departmental collaboration mechanisms to promote innovation and optimization. Fourthly, in Technology and Interaction Innovation, introduce intelligent interaction technologies, such as voice recognition and gesture control; optimize interface response speed and interaction fluency; and provide personalized interface customization and recommendation functions. Fifthly, for Continuous Iteration and Optimization, establish user feedback mechanisms; conduct regular usability testing; and continuously iterate system design based on user feedback. Sixthly, concerning Performance and Cost Balance, control system development costs while enhancing user experience, and ensure the feasibility and economy of technical solutions. Lastly, for Quality Assurance, conduct rigorous performance testing and quality inspection, and verify system stability, security, and user experience consistency. Through this comprehensive and systematic approach, the human-computer interaction quality of automotive infotainment systems can be significantly improved, creating a more intelligent, convenient, and personalized driving experience for users. |
| PMC  BS-F:65.11 BS-R:66.06 BS-P:64.20 GLEU:12.36 METEOR:27.06 BLEU-4:7.85 ROUGE-2:4.32 ROUGE-L:11.17 | Based on a comprehensive analysis of four sub-tasks, I will provide a cross-domain, systematic solution for integrating User Experience (UX) in automotive infotainment systems. The Comprehensive Solution Framework includes: 1. Global Design Philosophy: Core Objective is to create a user-centric, safe, efficient, intelligent, and personalized in-vehicle interaction system. 2. Cross-Domain Collaborative Mechanism: Key Collaboration Points are to establish a PLM digital collaboration platform, build a closed-loop system integrating design, manufacturing, supply chain, and quality, and enable real-time demand tracking and rapid response. 3. Core Elements of User Experience Design: For Interaction Interface, this includes simplified interface hierarchy, intuitive graphical design, safety-first operational logic, and millisecond-level response speed. For Ergonomics, it involves precise adaptation of screen size and position, support for multi-modal interaction (voice/gesture/touch), and personalized configuration and intelligent memory. 4. Technological Innovation Path: Key Technologies are AR navigation display, multi-device interconnection, AI intelligent recommendation, blockchain traceability, and machine vision inspection. 5. Manufacturing and Quality Assurance: Manufacturing Processes include five-axis CNC precision machining, high-performance composite materials, nano surface treatment, and intelligent inspection systems. Quality Control involves multi-scenario simulation testing, AI intelligent assessment, user behavior big data analysis, and a continuous iterative optimization mechanism. 6. Supply Chain Strategy: Supplier Selection requires strict technical capability assessment, multi-sourcing, and long-term strategic cooperation. Cost Control involves economies of scale in procurement, lean production, and cost reduction through technological innovation. 7. Implementation Path and Investment: Suggested Investment includes 5-8 million dollars for a digital platform, 2-3 million dollars for supplier capability enhancement, and an annual budget of 1-1.5 million dollars for continuous optimization. Expected Outcomes are a 30% improvement in supply chain response speed, a 15-20% reduction in quality costs, and a 25% increase in UX design implementation efficiency. 8. Risk Management: Key Risk Points are standardization of technology, cross-departmental collaboration, and digital transformation. Mitigation Strategies include phased implementation, continuous iteration, special project team building, and regular performance evaluation. 9. Continuous Optimization Mechanism: Optimization Paths are real-time user feedback collection, agile development methodology, cross-team collaboration, and rapid iteration cycles. In conclusion, this comprehensive solution, through systematic thinking, deeply embeds the user experience concept into the entire lifecycle of automotive infotainment systems, achieving a perfect balance between technology, user needs, and business value. |
| MedAgents  BS-F:65.94 BS-R:66.28 BS-P:65.61 GLEU:14.60 METEOR:24.72 BLEU-4:9.21 ROUGE-2:5.46 ROUGE-L:12.48 | A Comprehensive Solution for Automotive Infotainment System UX Design involves an Overall Strategic Framework with: 1. A User-Centric Collaborative Design Model, including establishing a cross-departmental UX design committee, regular user experience workshops, and a multi-level user feedback loop. 2. Balancing Technological Innovation and User Needs, with core principles that technology serves user experience, simplifies complex interaction logic, and ensures system intuitiveness and ease of use. Specific Implementation Paths include: I. User Research and Scenario Analysis Phase, involving user persona construction (segmenting users, analyzing scenario needs, user behavior database) and scenario simulation and experience testing (virtual simulation, multi-scenario validation, collecting user data and emotional feedback). II. Technology Integration Strategy, covering intelligent interaction technologies (multi-modal systems, context-aware tech, NLP, personalized algorithms) and hardware/software synergy (standardized interfaces, modular architecture, flexible framework). III. Quality Assurance Mechanism, with performance testing (extreme environment adaptability, long-term reliability, security/privacy testing) and continuous iteration and optimization (rapid feedback response, regular performance evaluation, agile development). Key Success Factors are seamless cross-departmental collaboration, a user-centric design philosophy, balancing innovation with practicality, and continuous learning with rapid iteration. Expected Outcomes include improved user satisfaction, reduced system complexity, enhanced brand competitiveness, and a differentiated user experience. Through this comprehensive strategic framework, while maintaining technological leadership, we will create a more intelligent, intuitive, and personalized in-vehicle interaction experience for users. |
| Debate  BS-F:65.36 BS-R:66.43 BS-P:64.33 GLEU:12.84 METEOR:33.59 BLEU-4:8.56 ROUGE-2:5.44 ROUGE-L:16.19 | As an expert in automotive design, I strongly agree with the in-depth analysis of the previous experts. I will provide more detailed and forward-looking insights from the perspective of User Experience (UX) design. The Comprehensive Innovation Strategy for Infotainment System UX Design includes: 1. Perceptual Intelligence in User Experience, which involves contextual intelligent perception through multi-sensor fusion, dynamic adjustment of cognitive load based on driving complexity, and emotional interaction design using emotional computing. 2. Innovative Paths in Interaction Design, featuring multi-modal intelligent interaction (voice, gesture, eye tracking, haptic feedback) and contextual predictive services (scenario prediction, intelligent recommendations). 3. Deep Integration of Human-centric Technology, encompassing privacy and security (data anonymization, user-controllable sharing, transparent algorithms) and inclusive design (age-friendly, accessible for disabilities, multi-language adaptability). 4. Seamless Ecosystem Connectivity, covering in-vehicle (autonomous driving collaboration, vehicle health monitoring), mobile (smartphone sync, wearable data integration), and cloud service ecosystems (real-time navigation, personalized services, remote diagnostics). 5. Agile Mechanism for Continuous Innovation, including user feedback loops, open development platforms, interdisciplinary R&D, and rapid iteration. 6. Future Technology Foresight, such as immersive interaction (AR navigation, VR entertainment), autonomous learning systems (AI adaptation, dynamic preference learning), and biometric technology (multi-factor authentication, personalized configurations). A key insight is that modern infotainment systems are intelligent interaction platforms requiring a balance between innovation, UX, and safety. Future systems will be mobile ecosystem portals with high intelligence and emotional perception. This multi-dimensional approach will greatly enhance UX quality and found future mobility ecosystems. |
| CoA  BS-F:65.56 BS-R:66.64 BS-P:64.71 GLEU:10.89 METEOR:28.42 BLEU-4:7.47 ROUGE-2:4.30 ROUGE-L:11.92 | Based on detailed suggestions from experts in various professional segments, I will provide an integrated, collaborative, full value-chain solution for an intelligent infotainment system from a senior management perspective. The Intelligent Adaptive Interaction Infotainment System - Comprehensive Management Plan outlines: I. Strategic Vision: To create a future-oriented intelligent in-vehicle interaction system, establishing new standards for automotive infotainment systems through user experience, technological innovation, and industrial collaboration. II. Overall Design Principles: 1. User-centric. 2. Technology innovation-driven. 3. Full value-chain collaboration. 4. Sustainable development. III. Key Integration Strategies: 1. Synergy between User Experience and Technological Innovation involves deep integration of personalized AI algorithms and interaction design, balancing innovation and usability in multi-modal interaction technologies, and dynamic adaptive interface design. 2. Synergy between Supply Chain and Production Manufacturing includes modular and standardized design, flexible production lines, agile supplier management, and a balance between cost and innovation. 3. Full Lifecycle Quality Management Plan encompasses proactive quality management, digital twin technology, continuous improvement mechanisms, and data-driven optimization. IV. Technology Roadmap: Phase 1 (6-12 months) involves building a prototype system, user testing and iteration, preliminary supplier assessment, and production line proof-of-concept. Phase 2 (12-24 months) includes system iteration and optimization, preparation for mass production, establishment of a supply chain network, and improvement of the quality management system. Phase 3 (24-36 months) focuses on full commercialization, international market expansion, continuous technological iteration, and ecosystem construction. V. Key Integration Mechanisms: 1. A Cross-departmental Collaboration Platform features regular technical roadmap discussions, agile decision-making mechanisms, and innovation incentive mechanisms. 2. Risk Management includes multi-scenario simulation, alternative technology routes, and rapid response mechanisms. 3. Balance between Innovation and Cost is achieved through cost reduction via modular design, economies of scale in procurement, and technological iteration. VI. Key Performance Indicators (KPIs) for User Experience are user satisfaction >95%, personalized recommendation accuracy >90%, and interaction response time <100ms. For Technological Innovation, KPIs include >10 patent applications/year, R&D investment ratio >5%, and technology iteration cycle <12 months. For Cost Control, targets are system cost 15% lower than similar products, production efficiency increased by 20%, and energy consumption reduced by 15%. For Quality Management, goals are product yield rate >99.5%, system reliability >99.99%, and defect rate <50 ppm. VII. Sustainable Development Commitment includes green production, low-carbon technologies, social responsibility, and technology inclusiveness. VIII. Implementation Guarantees are continuous high-level support, dedicated funding, talent development programs, and an open innovation ecosystem. In summary, this comprehensive plan, through deep collaboration in user experience, technological innovation, supply chain management, production manufacturing, and quality inspection, constructs a future-oriented intelligent in-vehicle infotainment system solution. The plan balances innovation, feasibility, cost, and quality, providing a systematic path for the development of automotive intelligent interaction systems. |

Figure 11: Case study on the Automotive dataset. Passages aligning with the reference answer are highlighted in the same color. The left side of the figure lists the scores for each baseline.

| Question | In toxicological research, how can high-throughput screening technology be used for the early identification and assessment of potential drug toxicity risks? |
| --- | --- |
| reference | In toxicological research, utilizing High-Throughput Screening (HTS) technology for the early identification and assessment of potential drug toxicity risks is an indispensable part of modern drug development. The following is a detailed answer to this question: 1. Principles and Applications of High-Throughput Screening Technology Principles: HTS technology, through automated equipment and miniaturized experiments, enables rapid screening of a large number of compounds in a short time to detect their effects on specific biomarkers or cellular functions. Applications: In toxicological research, HTS is primarily used to assess the potential risks of drugs in terms of cytotoxicity, genotoxicity, metabolic toxicity, etc. 2. Strategies for Early Toxicity Identification 2.1 Target-Specific Screening - Receptor/Enzyme Activity Detection: Evaluate the interference of drugs on key biological processes by utilizing changes in the activity of specific receptors or enzymes. - Cell Signaling Pathway Analysis: Predict potential toxicity mechanisms by detecting the effects of drugs on cell signaling pathways. 2.2 Cytotoxicity Screening - Cell Viability Assay: Use reagents such as MTT, Alamar Blue, etc., to assess the impact of drugs on cell proliferation and survival. - Cell Morphology Analysis: Detect changes in cell morphology through microscopic observation or automated image analysis systems. 2.3 Genotoxicity Screening - DNA Damage Detection: Utilize the Comet Assay or γH2AX focus formation assay to evaluate the damaging effects of drugs on DNA. - Gene Mutation Detection: Use the Ames test or micronucleus test to detect drug-induced gene mutations. 2.4 Metabolic Toxicity Screening - Hepatocyte Toxicity Detection: Utilize hepatocyte cell lines (such as HepG2) to assess the interference of drugs on liver metabolic functions. - Mitochondrial Function Detection: Evaluate drug-induced damage to mitochondrial function by measuring ATP generation or mitochondrial membrane potential. 3. Application of Latest Technological Methods 3.1 High-Content Screening (HCS) - Multi-Parameter Analysis: Provides more comprehensive toxicity information by combining cell morphology, fluorescent labeling, and image analysis. - Time-Resolved Imaging: Dynamically monitors cellular changes under drug action, revealing the temporal characteristics of toxic effects. 3.2 Omics Technologies - Genomics: Identify drug-induced gene expression changes through whole-genome expression analysis. - Metabolomics: Analyze drug-induced changes in metabolites to reveal potential metabolic toxicity. 3.3 Machine Learning and Artificial Intelligence - Data Mining: Utilize machine learning algorithms to extract toxicity-related features from large HTS datasets. - Predictive Model Construction: Build AI models for predicting drug toxicity based on existing data, improving screening accuracy and efficiency. 4. Practical Guidance Significance 4.1 Optimization of Screening Processes - Standardized Operations: Establish standardized HTS operating procedures to ensure the reliability and reproducibility of experimental results. - Quality Control: Strictly control experimental conditions to reduce inter-batch variability. 4.2 Data Integration and Analysis - Multi-Dimensional Data Integration: Combine HTS data with in vitro and in vivo experimental data for comprehensive risk assessment. - Bioinformatics Analysis: Utilize bioinformatics tools for in-depth analysis of toxicity mechanisms. 4.3 Risk Assessment and Management - Early Elimination of High-Risk Drugs: Timely eliminate candidate drugs with potential toxicity based on HTS results, reducing R&D costs. - Toxicity Mechanism Research: Conduct in-depth research on identified toxicity signals to provide guidance for subsequent drug design. 5. Industry Trends and Outlook - Personalized Toxicology: Conduct personalized toxicity risk assessment by combining patient genomic information. - Multidisciplinary Integration: Integrate knowledge from multiple disciplines such as chemistry, biology, and informatics to enhance the comprehensiveness and accuracy of toxicity screening. - Green Toxicology: Develop environmentally friendly and efficient toxicity screening methods, reducing the use of experimental animals. In summary, utilizing HTS technology for early identification and assessment of potential drug toxicity risks requires the comprehensive application of various technical means, optimization of experimental processes, and integration with the latest research progress and industry trends to achieve efficient and accurate toxicity assessment, providing strong support for drug development. |
| Cochain 🏅

BS-F:**76.04**
BS-R:**76.86**
BS-P:**75.24**
GLEU:**24.63**
METEOR:**36.10**
BLEU-4:**18.47**
ROUGE-2:**13.09**
ROUGE-L:**24.35** | I will provide you with a detailed analysis of the application of high-throughput screening technology in early risk assessment in toxicological research within the pharmaceutical workflow. Application of High-Throughput Screening Technology in Early Risk Assessment of Drug Toxicology I. Technology Overview High-Throughput Screening (HTS) is a key technology in modern drug development for rapidly assessing the safety and toxicity of compounds. Through automated and parallelized methods, it allows for systematic toxicity risk assessment of a large number of compounds in a short period. II. Key Technological Strategies 1. Cell-Level Toxicity Screening - Establishment of multiple cell models: Utilizing human-derived cell lines, primary cells, etc., to simulate the physiological environments of different organs and tissues. - Key detection indicators: - Cell viability - Apoptosis - Oxidative stress response - Mitochondrial function 2. Molecular-Level Risk Assessment - Genomic screening: - Detection of gene expression changes - Assessment of potential genotoxicity - Analysis of cellular stress response pathways - Proteomic analysis: - Protein interaction networks - Changes in signal transduction pathways - Key target toxicity assessment 3. Computational Toxicology Methods - Machine learning algorithms: Constructing toxicity prediction models - Quantitative Structure-Activity Relationship (QSAR) analysis - Molecular docking simulations - Bioinformatics risk prediction III. Technology Implementation Process 1. Preliminary Compound Library Construction - Collection and organization of candidate compounds - Standardized and structured data processing 2. High-Throughput Screening Stage - Multi-dimensional, multi-scale parallel detection - Automated experimental platforms - Real-time data acquisition and analysis 3. Risk Stratification and Assessment - Establishment of risk grading standards - Formulation of elimination and optimization strategies IV. Key Technological Challenges - Accuracy of simulating complex biological systems - Data standardization and interpretation - Reliability of computational models - Ethical and regulatory compliance V. Future Development Trends - AI-assisted toxicity prediction - Organoid technology - Personalized toxicological assessment - Interdisciplinary collaborative innovation Conclusion High-throughput screening technology, through multi-dimensional and high-efficiency methods, significantly enhances the ability to identify toxicity risks in the early stages of drug development, making it an indispensable key technological link in the modern pharmaceutical workflow. This systematic analysis report comprehensively elaborates on the application of high-throughput screening technology in the early risk assessment of drug toxicology from multiple dimensions such as technology, methods, processes, and future trends. Through collaborative innovation and interdisciplinary integration, this technology is reshaping the risk management paradigm in pharmaceutical R&D. |
| PMC

BS-F:65.68
BS-R:66.26
BS-P:65.12
GLEU:14.17
METEOR:23.61
BLEU-4:6.67
ROUGE-2:4.96
ROUGE-L:14.33 | Based on the results of the three sub-tasks, I will provide a comprehensive, cross-domain collaborative integrated solution for drug toxicity risk management: [Integrated Solution: Intelligent Drug Full-Lifecycle Toxicity Risk Management System] I. Overall Architecture 1. Core Concept: End-to-end intelligent risk warning and management 2. Key Technologies: Multi-dimensional high-throughput screening + AI risk prediction 3. System Goal: Achieve safety and controllability throughout the drug's full lifecycle II. Technological Innovation Pathways 1. R&D Stage (Toxicological Screening) - Build multi-target cell and molecular level screening platforms - Apply human-derived cell lines and 3D organ-on-a-chip - Integrate computational toxicology models - Key Technologies: * High-content imaging analysis * Gene expression microarrays * Proteomics detection 2. Manufacturing Stage - Intelligent real-time monitoring platform - Embed computational toxicology models - Establish multi-dimensional risk warning system - Key Strategies: * High-throughput online testing * Intelligent equipment transformation * Ensure GMP compliance 3. Distribution and Marketing Stage - Digital risk traceability platform - Machine learning risk classification system - Blockchain technology for tracking - Key Initiatives: * Real-time risk alerts * Temperature and environmental monitoring * Precise product recall mechanism III. Core Technology Integration 1. Data-Driven Risk Assessment Model - Establish dose-toxicity correlation models - Assess long-term and cumulative toxicity risks - Continuously update risk prediction algorithms 2. Intelligent Risk Management Platform - Cross-departmental collaboration mechanism - Real-time data analysis - Dynamic risk file management… |
| MedAgents

BS-F:65.23
BS-R:65.60
BS-P:64.89
GLEU:14.51
METEOR:23.59
BLEU-4:7.26
ROUGE-2:4.18
ROUGE-L:14.08 | Based on the application of High-Throughput Screening (HTS) technology in drug toxicology research, as a senior manager in the pharmaceutical industry, I propose the following collaborative plan for the entire workflow: Overall Strategic Plan: Constructing an \"Intelligent-Integrated-Dynamic\" Drug Safety Risk Management Ecosystem I. Organizational Structure and Collaborative Mechanisms 1. Establishment of a Cross-Departmental \"Intelligent Drug Safety Risk Assessment Center\" - Composition: Professionals in toxicology, bioinformatics, clinical research, AI algorithms, production quality, etc. - Establish a flat, matrix-style management model - Hold regular cross-departmental technical collaboration meetings II. Technology Platform Construction 1. Multi-Dimensional HTS Technology Platform - High-throughput cell culture screening platform - Genomics/proteomics analysis platform - AI-assisted computational toxicology model platform 2. Key Technology Breakthrough Directions - Improve data processing accuracy - Develop more precise predictive models - Integrate machine learning algorithms III. Risk Management Process Re-engineering 1. Early R&D Stage - Establish a multi-parameter, multi-dimensional risk assessment indicator system - Implement an \"early-continuous-dynamic\" risk identification mechanism - Embed HTS data into the R&D decision-making process 2. Production and Quality Control - Formulate dynamic safety standards based on HTS data - Establish a real-time risk monitoring and warning system - Promote intelligent quality management 3. Market and Clinical Feedback - Construct a drug safety big data platform - Achieve closed-loop feedback between clinical data and HTS data - Continuously optimize risk assessment models IV. Resource Allocation and Investment 1. Technology R&D Investment - Allocate 20% of the annual R&D budget for HTS technology innovation - Establish an innovation incentive mechanism for research personnel 2. Talent Development - Establish joint talent development programs with universities and research institutions - Organize regular interdisciplinary technical exchanges V. … |
| Debate

BS-F:68.24
BS-R:67.79
BS-P:68.70
GLEU:12.12
METEOR:19.42
BLEU-4:3.05
ROUGE-2:7.04
ROUGE-L:15.34 | As a pharmaceutical drug R&D expert, I will provide an in-depth analysis and supplement from a professional perspective on the application of high-throughput screening technology in toxicological research: 1. Core Advantages of High-Throughput Screening Technology Technological Innovation Dimensions: - Multi-dimensional parallel detection capabilities - Rapid and precise risk identification - Significant reduction in R&D costs and time 2. Advanced Technology Platform Expansion Molecular and Cellular Level Integration Technologies: - High-content imaging cell analysis systems - Single-cell transcriptomics screening - Microfluidic chip toxicity detection - Electrophysiological function assessment technology 3. Precise Risk Assessment Methods In-depth Molecular Toxicology Analysis: - Epigenetic change detection - Panoramic mapping of cellular stress responses - Potential immunogenicity risk assessment - Endocrine disruptor identification 4. AI-Empowered Toxicology Intelligent Prediction Technologies: - Deep learning toxicity risk models - Cross-species toxicity extrapolation algorithms - Molecular structure-toxicity relationship prediction - Dynamic risk scoring systems 5. Cutting-Edge Integrated Innovation Strategies - Multi-omics data fusion - Real-time dynamic risk monitoring - Personalized toxicity risk assessment - Balancing ethics and safety Key Technological Features: - Ultra-high throughput - Multi-dimensional detection - Intelligent prediction - Precise risk assessment Through this systematic and intelligent approach, the accuracy, efficiency, and comprehensiveness of early drug safety assessment can be significantly improved. |
| CoA

BS-F:63.75
BS-R:63.84
BS-P:63.66
GLEU:14.86
METEOR:25.17
BLEU-4:8.10
ROUGE-2:3.71
ROUGE-L:14.21 | Based on the detailed suggestions from experts in each segment, I will provide an integrated collaborative plan for the entire workflow, ensuring the efficient implementation of high-throughput toxicity screening technology throughout the drug R&D, production, and sales processes. Integrated Pharmaceutical Workflow Collaborative Plan: High-Throughput Toxicity Screening and Risk Management Strategies I. Overall Collaborative Governance Framework 1. Cross-Departmental Integrated Management Committee - Composition: • Head of R&D • Director of Production • Head of Sales & Marketing • Quality Compliance Officer • Chief Scientific Officer 2. Collaborative Goals - Establish an end-to-end innovative drug development ecosystem - Ensure seamless collaboration across R&D, production, and sales - Continuously optimize risk management mechanisms II. Key Collaborative Mechanisms 1. Information Sharing Platform - Establish a unified data management system - Real-time sharing of toxicity screening, production, and market data - Ensure multi-departmental collaborative decision-making 2. Risk Assessment and Management - Establish dynamic risk assessment models - Multi-dimensional risk warning mechanism - Rapid response and corrective measures III. …VI. Risk Management and Compliance 1. Regulatory Compliance - Strict adherence to FDA/EMA guidelines - Establish a comprehensive compliance audit system - Maintain transparency and traceability 2. Ethics and Safety - Establish an ethics review committee - Ensure ethical standards in R&D and production - Protect the rights of subjects and patients VII. Innovation and Continuous Optimization 1. Technology Iteration - Continuously invest in new technology R&D - Focus on the application of AI and big data in toxicity screening - Establish a technology radar to track cutting-edge innovations 2. Talent Development - Establish interdisciplinary talent development programs - Encourage internal innovation - Attract high-end talent Conclusion: This integrated plan, by establishing strong collaborative mechanisms, achieves seamless connection across R&D, production, and sales, ensuring the efficient application of high-throughput toxicity screening technology throughout the drug's full lifecycle. Through continuous optimization and innovation, we will establish a dynamic, agile, and risk-controlled drug development ecosystem. |

Figure 12: Case study on the Pharmaceutical dataset. Passages aligning with the reference answer are highlighted in the same color. The left side of the figure lists the scores for each baseline.

| Question | How can e-commerce platforms leverage procurement synergies to achieve cross-category and cross-regional procurement integration? |
|---|---|
| reference | E-commerce platforms can adopt the following strategies and methods to leverage procurement synergies and achieve cross-category and cross-regional procurement integration: 1. Data Analysis and Demand Forecasting: - Utilize big data analytics technology to collect and analyze market demand, sales data, inventory status, and other information across different categories and regions. - Based on data analysis results, predict sales trends for a future period to provide a basis for procurement decisions. 2. Supply Chain Integration: - Optimize the supply chain structure, integrate supplier resources, and establish a cross-category, cross-regional supply chain collaboration platform. - Establish long-term cooperative relationships with suppliers to achieve resource sharing and reduce procurement costs. 3. Centralized Procurement: - Centralize the processing of procurement needs for different categories and regions to achieve volume-based pricing and enhance bargaining power - Implement unified procurement strategies to avoid duplicate purchasing and resource waste. 4. Standardized Processes: - Establish standardized procurement processes and operating procedures to ensure the standardization and consistency of procurement activities. - Optimize procurement processes to improve procurement efficiency and reduce procurement costs. 5. Cross-Category Collaboration: - Integrate procurement needs across different categories to achieve economies of scale - Consider inter-category correlations in procurement decisions for comprehensive evaluation. 6. Regional Collaboration: - Reasonably allocate procurement resources based on sales conditions in different regions to achieve inter-regional procurement collaboration. - Establish regional procurement centers to achieve centralized management of procurement activities. 7. Information Sharing and Communication: - Establish an internal information sharing mechanism within the e-commerce platform to ensure information flow among procurement, inventory, sales, and other departments. - Strengthen communication with suppliers to promptly understand market changes and adjust procurement strategies. 8. Supply Chain Finance: - Utilize supply chain finance tools to provide financing support for suppliers and reduce procurement costs. - Optimize supply chain capital flow through financial means to improve the overall operational efficiency of the supply chain. 9. Continuous Optimization: - Regularly evaluate procurement synergies, analyze existing problems and shortcomings, and continuously optimize procurement strategies. - Introduce advanced procurement management tools and technologies to improve procurement synergies. Through the above measures, e-commerce platforms can effectively achieve cross-category and cross-regional procurement integration, improve procurement efficiency, reduce procurement costs, and thereby enhance market competitiveness. |
| Cochain

BS-F:**77.22**
BS-R:**77.22**
BS-P:**77.22**
GLEU:**23.26**
METEOR:34.58
BLEU-4:**16.18**
ROUGE-2:**19.39**
ROUGE-L:**27.43** | From the collaborative perspective of the e-commerce workflow, I will comprehensively answer from multiple dimensions how e-commerce platforms can achieve cross-category and cross-regional procurement integration: 1. Data-Driven Procurement Collaboration Strategy - Utilize big data analytics and demand forecasting technology to build a cross-category, cross-regional intelligent procurement model. - Accurately identify consumer demand in different regions and categories through user behavior data and market trend analysis. - Establish a unified data analysis platform to achieve scientific and precise procurement decision-making. 2. Supplier Ecosystem Collaboration - Build a diversified and standardized supplier management system. - Establish a cross-category, cross-regional supplier evaluation and admission mechanism. - Promote collaborative development among suppliers in terms of compliance, social responsibility, and sustainability. 3. Key Measures for Procurement Integration - Establish a centralized procurement platform to achieve economies of scale in purchasing. - Reduce logistics and procurement costs through regional collaboration. - Build flexible procurement strategies to quickly respond to different regional market demands. 4. Risk Management and Compliance Control - Formulate unified procurement compliance policies. - Ensure the legality and transparency of the procurement process. 5. Technology Empowerment and Innovation - Use AI and machine learning to optimize procurement forecasting models. - Build an intelligent supply chain management system. - Achieve digitalization and intelligence throughout the entire procurement process. Through the above strategies, e-commerce platforms can achieve procurement synergies, improving overall operational efficiency and market competitiveness. |
| PMC

BS-F:72.56
BS-R:72.53
BS-P:72.59
GLEU:18.77
METEOR:**38.60**
BLEU-4:6.67
ROUGE-2:4.96
ROUGE-L:14.33 | Based on the comprehensive results of four sub-tasks, I will provide a cross-domain, systematic e-commerce platform procurement collaboration solution: E-commerce Platform Cross-Category and Cross-Regional Procurement Collaboration Integrated Solution I. Strategic Architecture and Core Objectives 1. Overall Strategic Positioning - Build an \"Intelligent + Collaborative\" full-link procurement ecosystem. - Core Objectives: Enhance procurement efficiency, reduce costs, and strengthen supply chain resilience. - Achieve cross-category and cross-regional collaborative optimization. II. System Architecture and Key Elements 1. Technology Foundation - Big data analytics platform - AI and machine learning algorithms - Blockchain technology - IoT device network - Cloud computing infrastructure 2. System Functional Modules (1) Data Mid-platform - Cross-departmental, cross-category data integration - Unified data standards - Real-time data processing (2) Intelligent Decision Layer - Machine learning correlation analysis - Collaborative filtering recommendation models - Dynamic strategy adjustment algorithms (3) Execution Layer - Refined procurement management - Supplier capability profiling - Regionalized procurement strategies 3. Core Collaborative Mechanisms - Centralized procurement platform - Integration of supplier resource pools - Unified procurement standards and processes III. Key Implementation Strategies 1. Procurement Collaboration Optimization (1) Category Cross-Collaboration - Identify categories with similar supply chain characteristics - Consolidate bargaining power - Reduce overall procurement costs (2) Supplier Resource Management - Establish a multi-category supplier evaluation mechanism - Develop strategic partnerships - Promote supplier capability enhancement 2. Regional Supply Chain Network (1) Layout of Regional Procurement Centers - Based on transportation hubs and economically dense areas - Achieve localized procurement with national coordination - Reduce logistics costs (2) Regional Differentiated Management - Adjust strategies based on regional market characteristics - Flexibly respond to local supply chain challenges 3. Warehousing and Logistics Collaboration (1) Intelligent Warehousing Network - Precise positioning of regional warehousing centers - Intelligent cross-warehouse resource allocation - RFID and blockchain technology for tracking (2) Logistics Dispatch Optimization - AI algorithms for real-time route optimization - Cross-category order consolidation - Dynamic routing adjustments 4. User Experience Optimization (1) Personalized Recommendations - Machine learning algorithms for precise matching - Dynamic updates of user preferences - Expand product discovery range (2) Price Transparency - Directly convert procurement cost savings into user benefits - Long-term user price discount mechanisms 5. … |
| MedAgents

BS-F:71.00
BS-R:71.46
BS-P:70.54
GLEU:19.43
METEOR:32.70
BLEU-4:11.87
ROUGE-2:7.76
ROUGE-L:19.49 | An integrated solution for cross-category and cross-regional procurement collaboration on e-commerce platforms: [Overall Strategic Framework] I. Overall Collaborative Governance Model 1. Establish a \"Headquarters-Region-Category\" three-tier collaborative governance architecture - Headquarters level: Formulate overall strategy and standards - Regional level: Responsible for localized execution and optimization - Category level: Provide specialized procurement insights 2. Construct a Cross-Functional Collaborative Committee - Members include procurement, technology, operations, finance, etc. - Hold regular cross-departmental strategic coordination meetings - Establish rapid decision-making mechanisms [Technology Support System] II. Intelligent Procurement Collaboration Platform 1. Technology Architecture - Build a cloud-native distributed procurement management system - Integrate big data, AI algorithms, and blockchain technology - Ensure system real-time performance, security, and scalability 2. Core Functional Modules - Panoramic supplier profiling - Intelligent matching and recommendation - Dynamic price forecasting - Real-time risk monitoring [Supplier Management Mechanism] III. Standardized Supplier Ecosystem Governance 1. Admission and Evaluation System - Establish unified supplier evaluation standards - Construct a dynamic rating mechanism - Implement differentiated management strategies 2. Collaborative Incentive Mechanism - Establish a long-term strategic partnership program - Design a multi-dimensional performance evaluation system - Provide precise capability training and support [Operational Optimization Strategy] IV. Full-Link Cost and Efficiency Optimization 1. Procurement Cost Control - Establish a centralized procurement platform - Achieve economies of scale in bargaining - Promote category integration 2. Logistics and Warehousing Collaboration - Build a regional logistics center network - Optimize inventory management - Reduce warehousing and delivery costs [Risk Management] V. Multi-Dimensional Risk Prevention and Control 1. Supply Chain Resilience - Establish diversified supplier reserves - Formulate emergency plans - Construct a risk warning mechanism 2. Data Security - Implement strict data protection measures - Establish a compliance management framework - Protect business privacy [Implementation Path] VI. … |
| Debate

BS-F:72.56
BS-R:72.79
BS-P:72.32
GLEU:21.31
METEOR:35.04
BLEU-4:14.94
ROUGE-2:10.36
ROUGE-L:19.86 | As an expert in e-commerce merchandise procurement, I strongly agree with the systematic solution proposed by the previous expert. On this basis, I will further deepen and supplement the implementation path for achieving procurement synergies from a practical perspective. In-depth Practical Paths for E-commerce Platform Cross-Category and Cross-Regional Procurement Synergies: 1. Refined Collaborative Management (1) Dynamic Collaborative Matrix - Establish a cross-category, cross-regional procurement collaboration matrix - Formulate differentiated procurement strategies - Achieve dynamic optimization and allocation of procurement resources (2) Collaborative Performance Evaluation System - Construct multi-dimensional collaborative performance indicators - Establish cross-team, cross-category incentive mechanisms - Quantify collaborative effects and value creation 2. Data-Driven Precise Collaboration (1) Full-Link Data Integration - Integrate procurement, sales, and inventory data - Build a panoramic data insight platform - Achieve real-time data sharing and collaboration (2) Intelligent Prediction and Decision-Making - Develop machine learning-based demand forecasting models - Establish dynamic pricing and inventory optimization algorithms - Achieve precision and intelligence in procurement 3. Supplier Ecosystem Collaboration (1) Tiered Management - Construct a supplier tiered evaluation system - Establish differentiated cooperation mechanisms - Promote supplier capability enhancement (2) Co-creating Value Ecosystem - Build a supplier collaborative innovation platform - Promote product R&D and technology iteration - Build mutually beneficial long-term partnerships 4. Scenario-Based Collaborative Innovation (1) Cross-Border Integration - Explore cross-industry, cross-domain procurement collaboration - Discover emerging procurement scenarios and business models - Promote procurement model innovation (2) Flexible Supply Chain - Build an agile and resilient supply chain network - Establish rapid response mechanisms - Enhance supply chain resilience and risk resistance 5. Sustainable Development Collaboration (1) Green Procurement - Formulate green supply chain standards - Promote supplier sustainable development - Build a responsible procurement ecosystem (2) Social Responsibility - Establish a supplier social responsibility assessment mechanism - Promote social value creation in the supply chain - Balance economic benefits with social value Implementation Path: 1. Top-down strategic guidance 2. Continuous technological and managerial innovation 3. Open and shared ecosystem construction 4. Data-driven lean management Core Competencies: - Ecosystem collaboration capability - Data insight capability - Scenario innovation capability - Sustainable development capability Summary: Procurement synergy for e-commerce platforms is a systematic project that requires systematic promotion from multiple dimensions such as … |
| CoA

BS-F:71.62
BS-R:72.59
BS-P:70.67
GLEU:17.45
METEOR:37.22
BLEU-4:13.89
ROUGE-2:6.98
ROUGE-L:16.62 | Based on the detailed suggestions from experts in various segments of the e-commerce platform workflow, I will provide an integrated, collaborative final solution. This plan, from a strategic height, will balance multiple dimensions such as procurement, operations, logistics, and user experience to build a comprehensive and efficient e-commerce platform ecosystem. E-commerce Platform Full Workflow Collaborative Development Strategy Plan I. Strategic Vision Build an intelligent e-commerce platform ecosystem centered on data-driven insights, technological innovation, and user experience, achieving economies of scale, cost optimization, and differentiated competitive advantages through cross-segment collaboration. 2. Core Collaborative Mechanisms 2.1 Data Mid-platform and Intelligent Platform 1. Unified Data Governance - Construct a centralized data mid-platform - Integrate procurement, operations, logistics, and user data - Achieve real-time data sharing and analysis 2. AI Intelligent Decision System - Build cross-segment intelligent algorithms - Support decisions in procurement, inventory, pricing, recommendations, etc. - Achieve data-driven precise operations 2.2 Supply Chain Collaboration 1. Full-Link Supplier Management - Unified supplier evaluation standards - Establish a dynamic performance mechanism - Form a strategic partner ecosystem 2. Flexible Supply Chain - Rapid response to market demands - Support cross-category, cross-regional procurement - Optimize inventory and delivery efficiency 3. Key Collaborative Strategies 3.1 Procurement and Operations Collaboration 1. Data-Driven Merchandise Strategy - Based on user behavior and market insights - Formulate precise procurement plans - Optimize product structure and listing strategies 2. Promotion and Procurement Linkage - Cross-category, cross-regional combined promotions - Leverage procurement scale advantages - Increase average transaction value and conversion rates 3.2 Logistics and User Experience Collaboration 1. Personalized Delivery Services - Delivery strategies based on user profiles - Flexible delivery time choices - Real-time logistics transparency 2. Intelligent Warehousing Supports User Experience - Rapid order processing - Precise inventory management - Support innovative experiences like AR/VR 4. Technological Innovation Pathways 4.1 Core Technology Architecture 1. Microservices Architecture - Highly decoupled system design - Rapid iteration and flexible scalability - Support independent optimization of each module 2. Blockchain and Security Technology - Enhance supply chain transparency - Protect user data security - Establish a trustworthy business ecosystem 4.2 Artificial Intelligence Applications 1. Intelligent Recommendations and Personalization - Multi-dimensional user profiling - Real-time interest analysis - Dynamic personalized content 2. Predictive Analytics - Demand forecasting - Price optimization - Risk management 5. Organizational Capability Building… |

Figure 13: Case study on the E-commerce dataset. Passages aligning with the reference answer are highlighted in the same color. The left side of the figure lists the scores for each baseline.

## F    PROMPTS TEMPLATE

We provide the prompt templates for all multi-agent baselines presented in the paper, exemplified using the automotive business workflow.

### F.1    COCHAIN

We present a specific overview of the prompts template in the Cochain. As illustrated in Figure 14, the prompt template adopts a modular design comprising three core components: the knowledge module retrieved from the collaborative knowledge graph, the causal chain module, and the business workflow prompts module generated through prompts tree retrieval. In the template, the gray box represents fixed prompt content, while the colored boxes denote dynamic content that adapts based on user needs.

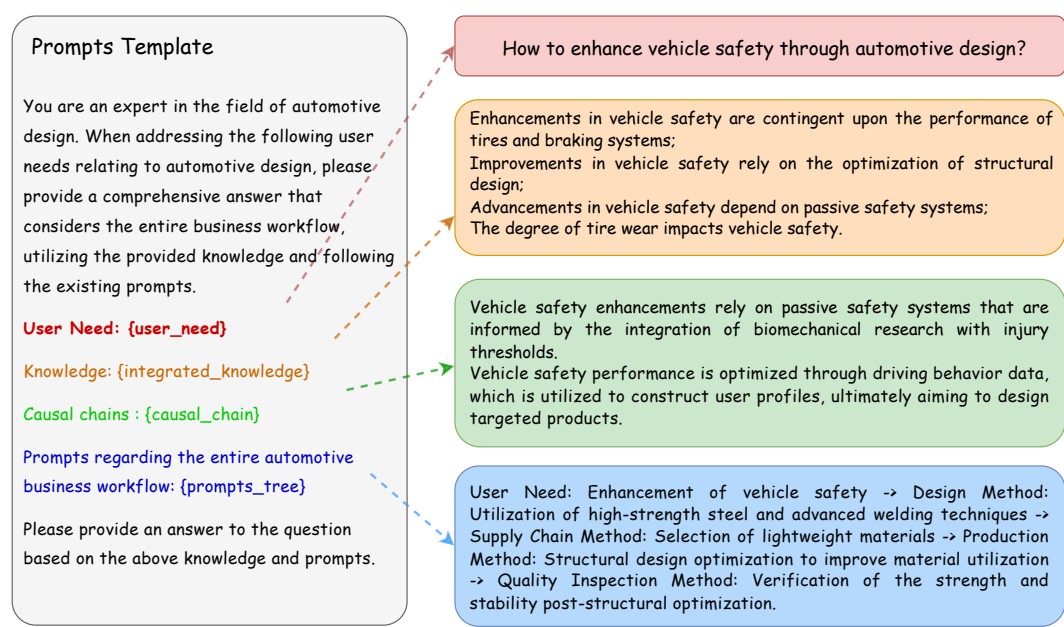

Figure 14: The prompts template of Cochain for final input to LLM.

## F.2 PMC

**Manager Agent**

**Role:** You are a senior project manager in the automotive industry, responsible for task decomposition and planning.

**Prompt:** Please analyze the following automotive-related problem: **{question}**

Your task is to:
1. Analyze which segments of the automotive value chain this problem involves (design, manufacturing, supply chain, quality control)
2. Decompose the problem into subtasks corresponding to different segments
3. Determine the dependencies between subtasks
4. Identify local constraints for each subtask and global constraints

Please output the task decomposition result in JSON format, including the following fields:
- main_task: Description of the main task
- global_constraints: List of global constraints
- sub_tasks: Dictionary of subtasks, where each subtask contains content (task description), domain (relevant area such as design/manufacturing/supply chain/quality control), local_constraints (constraints specific to this subtask), require_data (list of other subtasks this subtask depends on)

Figure 15: The manager agent prompts template of PMC.

**Supervisor Agent**

**Role:** You are a cross-domain coordination expert in the automotive industry, responsible for integrating solutions from different segments.

**Prompt:** You need to optimize the execution of the current subtask based on the results of the following prerequisite subtasks:

Current task: **{current_task}**
Results of prerequisite tasks: **{previous_tasks_results}**

Please reframe the current task to fully consider the results and constraints from prerequisite tasks, forming a more coordinated solution.

Figure 16: The supervisor agent prompts template of PMC.

**Deliverer Agent**

**Role:** You are a comprehensive solution expert in the automotive industry, responsible for integrating the outcomes of all subtasks.

**Prompt:** Based on the results of all the following subtasks, provide a comprehensive solution that satisfies global constraints:

Original problem: **{question}**
Subtask results: **{subtasks_results}**
Global constraints: **{global_constraints}**

Please provide a comprehensive, cross-domain collaborative solution that ensures coordination and consistency between all segments while meeting all constraints.

Figure 17: The deliverer agent prompts template of PMC.

**Executor Agent-Design Domain**

**Role:** You are an expert in the automotive design domain. You are familiar with automotive design principles, ergonomics, aesthetic design, and functional design.

**Prompt:** Based on the following subtask description, please provide professional automotive design analysis and solutions:

**{task_content}**

Please pay special attention to the following local constraints:
**{local_constraints}**

Please provide detailed design plans and considerations.

**Executor Agent-Supply Domain**

**Role:** You are an expert in the automotive supply chain domain. You are familiar with parts procurement, supplier management, inventory optimization, and logistics management.

**Prompt:** Based on the following subtask description, please provide professional automotive supply chain analysis and solutions:

**{task_content}**

Please pay special attention to the following local constraints:
**{local_constraints}**

Please provide detailed supply chain plans and considerations.

**Executor Agent-Manufacturing Domain**

**Role:** You are an expert in the automotive manufacturing domain. You are familiar with production processes, assembly line design, production efficiency optimization, and manufacturing technologies.

**Prompt:** Based on the following subtask description, please provide professional automotive manufacturing analysis and solutions:

**{task_content}**

Please pay special attention to the following local constraints:
**{local_constraints}**

Please provide detailed manufacturing plans and considerations

**Executor Agent-QC Domain**

**Role:** You are an expert in the automotive quality control domain. You are familiar with quality control methods, inspection techniques, quality standards, and quality improvement.

**Prompt:** Based on the following subtask description, please provide professional automotive quality inspection analysis and solutions:

**{task_content}**

Please pay special attention to the following local constraints:
**{local_constraints}**

Please provide detailed quality inspection plans and considerations.

Figure 18: The executor agent prompts template of PMC.

## F.3 MEDAGENTS

---

**Stage One: Expert Gathering**

**Role:** You are an automotive industry consultant specializing in classifying specific automotive design problems into particular domains of the automotive business workflow.

**Prompt:** You need to complete the following steps:

1. Carefully read the problem described in the following automotive design scenario:
   {question}

2. Based on the automotive design scenario in the problem, classify the issue into four different automotive engineering and business workflow sub-domains.

3. You should output in the following format:
   Professional domain: |

---

**Stage Two: Analysis Proposition**

**Role:** You are an automotive engineering expert in the **{domain}** . Based on your professional domain, you will examine and analyze problems presented in specific automotive design scenarios.

**Prompt:** Please carefully examine the automotive design scenario outlined in this problem:

**{question}**

Relying on your automotive engineering expertise, explain the described situation. Subsequently, identify and highlight the aspects of the problem that you consider most worthy of attention or most noteworthy. Pay particular attention to how this design impacts or is impacted by other segments of the automotive business workflow (such as manufacturing, supply chain, quality inspection, etc.).

---

**Stage Three: Report Summarization**

**Role:** You are an automotive engineering assistant, skilled at summarizing and synthesizing multiple reports from experts across various domains.

**Prompt:** Below are reports from different automotive engineering domain experts.
**{reports}**
You need to complete the following steps:

1. Carefully and comprehensively consider the following reports.
2. Extract key knowledge from the following reports.
3. Develop a comprehensive analysis based on this knowledge.
4. Your ultimate goal is to produce a refined and synthesized report based on the following inputs.

You should output in the following format:
Key Knowledge:
Comprehensive Analysis:

---

Figure 19: The prompt templates for stages 1 through 3 of MedAgents.

**Stage Four: Collaborative Consultation—Voting**

**Role:** You are an automotive engineering expert specializing in the **{domain}**.

**Prompt:** This is an automotive engineering report:

**{synthesized_report}**

As an automotive engineering expert specializing in **{domain}**, please carefully read the report and decide if your perspective aligns with this report. Please only respond with: [Yes or No].

**Stage Four: Collaborative Consultation—Modification Suggestion**

**Role:** You are an automotive engineering assistant tasked with incorporating expert feedback.

**Prompt:** This is a suggestion from an automotive engineering expert specializing in **{domain}**:

**{advice}**

Based on the above suggestion, please output your modified analysis in the following format:
Key Knowledge:
Comprehensive Analysis:

**Stage Five: Decision Making**

**Role:** You are a senior manager in the automotive industry, responsible for integrating opinions from various professional domains and making final decisions.

**Prompt:** Original problem: **{question}**

Below is a comprehensive report that has reached consensus through expert collaboration:
**{final_report}**

As a manager, please:
1. Integrate opinions and suggestions from experts across all segments
2. Resolve potential conflicts between different segments
3. Balance innovation, feasibility, cost, and quality factors
4. Propose a comprehensive and collaborative final solution

Please provide a final solution that incorporates perspectives from the entire value chain, ensuring all segments work in coordination to achieve optimal overall results.

Figure 20: The prompt templates for stages 4 and 5 of MedAgents.

## F.4 DEBATE

---

**Expert Initial Prompt**

**Role:** You are an expert in the automotive design domain. Based on your professional domain, you will examine and analyze problems presented in specific automotive design scenarios.

**Prompt:** Please carefully examine the automotive design scenario outlined in this problem:

**{question}**

Relying on your automotive design expertise, explain the described situation. Subsequently, identify and highlight the aspects of the problem that you consider most worthy of attention or most noteworthy. Pay particular attention to how this design impacts or is impacted by other segments of the automotive value chain (such as manufacturing, supply chain, quality inspection, etc.).

---

**Expert Short Debate Prompt Template**

**Role:** You are an expert in the automotive design domain participating in a collaborative problem-solving session.

**Prompt:** These are the responses from other automotive design experts to the problem:

**{other_answers}**

Based on the perspectives of other experts, please provide your updated response.

---

**Expert Long Debate Prompt Template**

**Role:** You are an expert in the automotive design domain participating in a collaborative problem-solving session.

**Prompt:** These are the responses from other automotive design experts to the problem:

**{other_answers}**

Using the perspectives of other experts as additional suggestions, please provide your updated response. Please pay particular attention to how the design solution can both meet functional requirements and consider the constraints and requirements of the entire automotive value chain (including manufacturing, supply chain, quality inspection, and other segments).

---

Figure 21: The prompts template of Debate for final input to LLM.

## F.5 CoA

**Working Agent-Wi**

**Role:** You are a professional expert in the {current segment} of the automotive value chain.

**Prompt:** {Input the professional knowledge and constraints of the current segment ci}

This is the summary and suggestions from the previous segment:
{Communication unit from the previous working agent (CUi-1)}

User question: {query question q}

You need to read the professional knowledge of the current segment and the summary from the previous segment (if available), then generate a comprehensive summary that incorporates both.

This summary will be used for analysis in subsequent segments and to ultimately answer the user's question.

As a {current segment} expert, you need to:
1. Analyze the feasibility of what was proposed in the previous segment
2. Provide supplementary suggestions or modifications from the {current segment} perspective
3. Pay special attention to {concerns specific to the current segment}
4. Ensure your suggestions form a synergy with the previous segment rather than simply negating it

Please generate a comprehensive summary with sufficient details for reference in subsequent segments:

**Management Agent-M**

**Role:** You are a management agent responsible for synthesizing expert insights from across the automotive value chain.

**Prompt:** User question: {query question q}

Below are the comprehensive suggestions after collaborative analysis by experts from various segments of the automotive value chain (design, manufacturing, supply chain, quality control):
{Final communication unit from the working agent CUI}

As a management agent, you need to:
1. Analyze the suggestions and insights provided by experts from each segment
2. Integrate these suggestions, balancing various considerations
3. Resolve potential conflicts and propose optimal compromise solutions
4. Generate a comprehensive and coordinated final answer

Please provide the final collaborative answer, ensuring optimal balance between design innovation, manufacturing feasibility, supply chain efficiency, and quality assurance:

Figure 22: The prompt templates for the worker agent and manager agent of CoA.

**Automotive Design Expert**

**Role:** You are an automotive design expert within the working agent framework.

**Prompt:** As an automotive design expert, you need to:
1. Propose innovative yet practical design solutions
2. Consider aesthetics, ergonomics, aerodynamics, and other relevant aspects
3. Pay special attention to user experience and design trends
4. Anticipate the potential impacts of your design on subsequent manufacturing, supply chain, and quality inspection processes

**Automotive Supply Chain Expert**

**Role:** You are an automotive supply chain expert within the working agent framework.

**Prompt:** As an automotive supply chain expert, you need to:
1. Evaluate the availability of components and materials
2. Analyze supply chain risks and cost structures
3. Pay special attention to supplier selection for key components
4. Propose suggestions that optimize the supply chain while meeting design and manufacturing requirements

**Automotive Manufacturing Expert**

**Role:** You are an automotive manufacturing expert within the working agent framework.

**Prompt:** As an automotive manufacturing expert, you need to:
1. Analyze the manufacturability of the design solutions
2. Evaluate manufacturing costs and production efficiency
3. Pay special attention to process workflows and manufacturing technology limitations
4. Propose suggestions that optimize manufacturing feasibility while maintaining design intent

**Automotive Quality Control Expert**

**Role:** You are an automotive quality control expert within the working agent framework.

**Prompt:** As an automotive quality control expert, you need to:
1. Evaluate the quality assurance feasibility of design and manufacturing plans
2. Analyze potential quality risk points
3. Pay special attention to safety, durability, and consistency testing methods
4. Propose quality control measures that consider design intent, manufacturing processes, and supply chain constraints

Figure 23: The prompt templates for each worker agent on the Automotive dataset.

