# OpenReview forum: "Connecting the Dots: A Chain-of-Collaboration Prompting Framework for LLM Agents"
_ICLR.cc/2026/Conference — ICLR 2026 Conference Withdrawn Submission_

### Official Review · Reviewer_YHtV · 2025-10-28

**Soundness:** 2
**Presentation:** 2
**Contribution:** 2
**Rating:** 2
**Confidence:** 4

**Summary:**

This manuscript proposes Cochain, a framework designed to enhance the reasoning ability of LLM-based agents while reducing token costs. While multi-agent systems (MAS) are generally more capable than single-agent systems, they often suffer from excessive token usage, high inference latency, and over-collaboration. To address these challenges, Cochain introduces a knowledge-graph-based (KG-based) framework that constructs a task-relevant knowledge graph from the training dataset. For a new query, Cochain identifies a causal chain within the KG that is most relevant to the query to guide reasoning. Empirical studies demonstrate the superiority of Cochain over baseline methods.

**Strengths:**

The paper addresses the efficiency challenge of multi-agent systems (MAS) in business workflows, which is an important and practically relevant problem.

**Weaknesses:**

1. Limited Literature Review. The authors claim that the inefficiency and failures (e.g., over-collaboration) of MAS have not been studied (Line 76), which motivates their work. However, recent studies [1–3] have already examined MAS failures arising from agentic communication and coordination issues. The related work section should be expanded to include these efforts.

2. Methodological Positioning. The authors state that Cochain integrates chain-of-thought reasoning into agentic systems to reduce token cost. However, the proposed framework is conceptually closer to knowledge graph construction and KG-based retrieval-augmented generation (RAG), rather than explicit chain-of-thought reasoning. The distinction should be clarified.

3. Causal Chain Construction. The manuscript claims that Cochain builds a causal chain to guide reasoning. However, the method described in Section 3.2 merely extracts the top-K relevant triples from the KG. The causal inference mechanism or reasoning process linking these triples is not clearly defined or justified.

References

[1] Multi-agent Architecture Search via Agentic Supernet. ICML 2025.

[2] Which Agent Causes Task Failures and When? On Automated Failure Attribution of LLM Multi-Agent Systems. ICML 2025.

[3] Why Do Multi-Agent LLM Systems Fail? arXiv:2503.13657.

**Questions:**

The authors are encouraged to address the weaknesses above.

---

### Official Review · Reviewer_2PGG · 2025-11-01

**Soundness:** 2
**Presentation:** 3
**Contribution:** 3
**Rating:** 4
**Confidence:** 3

**Summary:**

This paper introduces Cochain, a novel prompting framework designed to improve the collaborative reasoning of Large Language Model (LLM) agents, particularly for complex, multi-stage "business workflow" tasks. The authors identify a key trade-off between single agent and multiagent. Cochain aims to achieve the "effective collaboration" of a multi-agent system at the low cost of a single-agent system. It does this by having agents collaborate indirectly through two main components built in a one-time setup process

**Strengths:**

1. The paper clearly articulates the concepts of "under-collaboration" and "over-collaboration". Identifying and providing a solution for the high cost and diluted focus of multi-agent systems.
2. The evaluation metrics include text similarity metrics and human evaluation.
3. Paper visualization is great and help to understand the paper and it also provides the cost analysis in the appendix, making it practical.

**Weaknesses:**

1. it is unknown if Cochain generalize beyond “business workflows” to creative, open-ended reasoning tasks (e.g., scientific discovery, multi-hop QA).
2. The Prompts Tree seems to encode a static, linear workflow. How would Cochain adapt to a dynamic business process where, for example, the output of Stage 2 determines whether Stage 3 or Stage 4 comes next?
3. Addition to 2, the framework needs to be completely rebuilt when accommodating changes in the workflow structure.
4. The counterfactual generation and tacit variable modeling seem conceptually interesting, but the paper doesn’t show ablation isolating this component from the rest (e.g., KG + causal chain without counterfactual reasoning). It’s therefore unclear whether counterfactual augmentation materially improves performance or just inflates data diversity.

**Questions:**

1. How scalable is Cochain in terms of graph size and prompt tree depth? Are there any empirical bottlenecks beyond the small domain examples?
2. Can Cochain be applicable to creative, open-ended reasoning tasks (e.g., scientific discovery, multi-hop QA)?
3. Are there risks of reinforcing domain biases when constructing the collaborative knowledge graph from domain-specific agents?

---

### Official Review · Reviewer_YiA1 · 2025-11-01

**Soundness:** 4
**Presentation:** 4
**Contribution:** 3
**Rating:** 8
**Confidence:** 3

**Summary:**

This paper introduces Cochain, a prompting framework designed for large language model (LLM) agents. The framework integrates a knowledge graph to aggregate and utilize information across multiple stages of reasoning and maintains a structured prompt tree to retrieve contextually relevant knowledge for complex business workflows.

**Strengths:**

- The paper is clearly written and effectively uses visual aids to illustrate the framework and workflow.
- Evaluation is conducted on two benchmarks spanning four datasets, providing a broad empirical basis.
- Extensive experiments across multiple LLMs strengthen the generality of the findings.
- Both human and automatic evaluations are performed, offering complementary perspectives on performance.
- Ablation studies and efficiency analyses provide valuable insights into the contributions and trade-offs of different components within Cochain.

**Weaknesses:**

The framework’s increased reliance on the knowledge graph and prompt tree leads to higher resource consumption. A detailed discussion of this limitation, including potential mitigation strategies, would improve the paper’s completeness.

**Questions:**

What computational resources (e.g., hardware specifications, number and type of GPUs, inference settings) were used to conduct all experiments?

---

### Official Review · Reviewer_gaLJ · 2025-11-03

**Soundness:** 2
**Presentation:** 1
**Contribution:** 2
**Rating:** 2
**Confidence:** 2

**Summary:**

The paper introduces Cochain, a novel framework that enhances collaboration and reasoning among LLM agents working across multi-stage workflows. It aims to solve two opposite issues in LLM systems: under collaboration and over collaboration.

**Strengths:**

1. Integrates explicit knowledge (from input-output data) and tacit knowledge (from counterfactual reasoning).

**Weaknesses:**

1. The paper is very hard to understand. As an examples, Figure 1 is not clear at all. Why is some text in red? what are the different stages of business workflow? Even Fig 2 is complicated.
2. Many parts of the paper are unclear:
- In equation 1, what is $n_i$, $N$, what does $j$ represent is not mentioned.
- In equation 2, how is counterfactual generated? The paper mentions "We then input these questions into the relevant vertical domain agents to generate answers". What are these agents?
- Is the Prompt Tree construction done for the entire dataset as a pre-processing step? or is it done per scenario or done dynamically? What model is used to construct the Prompt Tree?
3. If the goal is answer user queries through effective collaboration of agents then shouldn't the metric used be accuracy of the responses rather than rouge and gleu?

**Questions:**

See above.

---

### Note · Authors · 2025-11-20

I have read and agree with the venue's withdrawal policy on behalf of myself and my co-authors.